# Information Theoretic Regularization for Learning Global Features by Sequential VAE

## Abstract

Sequential variational autoencoders (VAEs) with global latent variable $z$ have been studied for the purpose of disentangling the global features of data, which is useful in many downstream tasks. To assist the sequential VAEs further in obtaining meaningful $z$, an auxiliary loss that maximizes the mutual information (MI) between the observation and $z$ is often employed. However, by analyzing the sequential VAEs from the information theoretic perspective, we can claim that simply maximizing the MI encourages the latent variables to have redundant information and prevents the disentanglement of global and local features. Based on this analysis, we derive a novel regularization method that makes $z$ informative while encouraging the disentanglement. Specifically, the proposed method removes redundant information by minimizing the MI between $z$ and the local features by using adversarial training. In the experiments, we trained state-space and autoregressive model variants using speech and image datasets. The results indicate that the proposed method improves the performance of the downstream classification and data generation tasks, thereby supporting our information theoretic perspective in the learning of global representations.

## 1 Introduction

Uncovering the *global factors* of variation from high-dimensional data is a significant and relevant problem in representation learning (Bengio et al., 2013). For example, a global representation of images that presents only the identity of the objects and is invariant to the detailed texture would assist in downstream semi-supervised classification (Ma et al., 2019). In addition, the representation is known to be useful in the controlled generation of data. Obtaining the representation allows us to manipulate the voice of the speaker in speeches (Yingzhen & Mandt, 2018), or generate images that share similar global structures (e.g. the structure of objects) but varying details (Razavi et al., 2019).

Sequential variational autoencoders (VAEs) with a global latent variable $z$ have played an important role in the unsupervised learning of the global features. Specifically, we consider the sequential VAEs with a structured data generating process in which an observation $x$ at time $t$ (denoted as $x_t$) is generated from a global feature $z$ and local feature $s_t$. Then, the $z$ of such sequential VAEs can acquire only global information invariant to $t$. For example, Yingzhen & Mandt (2018) demonstrated that a disentangled sequential autoencoder (DSAE), which combines state-space models (SSMs) with a global latent variable $z$, can uncover the speaker information from speeches. Furthermore, Chen et al. (2017); Gulrajani et al. (2017) proposed a VAE with a PixelCNN decoder (denoted as PixelCNN-VAE), which combines autoregressive models (ARMs) and $z$. In both methods, the hidden state of the sequential model (either SSMs or ARMs) is designed to capture local information, while an additional latent variable $z$ captures global information.

Unfortunately, the design of the aforementioned structured data generating process alone is insufficient to uncover the global features in practice. A typical issue is that latent variable $z$ is ignored by a decoder (SSMs or ARMs) and becomes uninformative. This phenomenon occurs as follows: with expressive decoders, such as SSMs or ARMs, the additional latent variable $z$ cannot assist in improving the evidence lower bound (ELBO), which is the objective function of VAEs; therefore, the decoders will not use $z$ (Chen et al., 2017; Alemi et al., 2018). The phenomenon in which the latent variables are ignored is referred to as posterior collapse (PC). To alleviate this issue, several studies have proposed regularizing the mutual information (MI) between $x$ and $z$ to be large, e.g.,

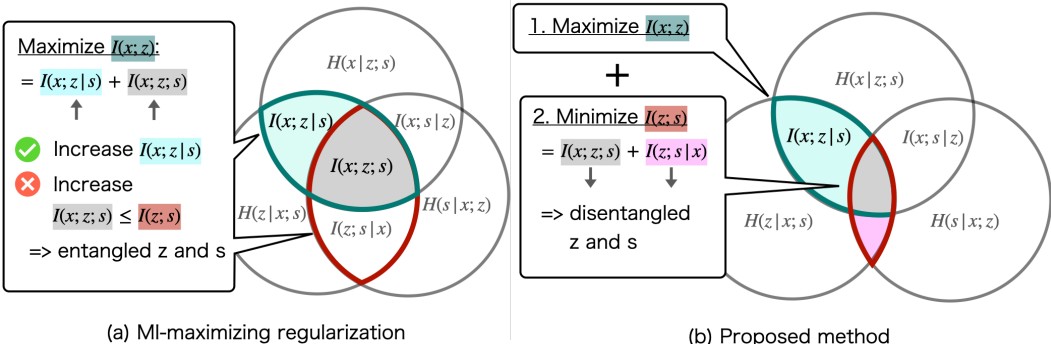

Figure 1: Comparison of (a) MI-maximizing regularization and (b) the proposed method, using a Venn diagram of information theoretic measures of $x$, $z$, and $s$.

using $\beta$-VAE (Alemi et al., 2018). A higher MI $I(x; z)$ indicates that $z$ has significant information regarding $x$; this regularization prevents $z$ from becoming uninformative.

In this paper, we further analyze the MI-maximization and claim that merely maximizing $I(x; z)$ is insufficient to uncover the global factors of variation. Figure 1-(a) summarizes the issue of the MI-maximization. As illustrated in the Venn diagram, the MI can be decomposed into $I(x; z) = I(x; z|s) + I(x; z; s)$. Maximizing the first term $I(x; z|s)$ is beneficial, as it measures the informativeness of $z$ about $x$ given a local feature $s$. However, maximizing the second term $I(x; z; s)$ might cause a negative effect, because it would also increase $I(z; s)$. In other words, maximizing $I(x; z)$ would encourage latent variables to have redundant information. For example, when $I(x; z)$ becomes so large that $z$ retains all (local and global) information of $x$, the downstream classification performance would be degrated. Also, when local variables still contain global information due to large $I(z; s)$, it becomes difficult to control speaker information in speeches using a DSAE. See Appendix A for empirical evidence that the MI-maximization increases $I(z; s)$, as discussed above.

Based on the analysis, we propose a new information theoretic regularization method for disentangling the global factors. Specifically, our method minimizes $I(z; s)$, in addition to maximizing $I(x; z)$ similar to prior work (Figure 1-(b)). As $I(z; s)$ measures the dependence between $z$ and $s$, our method encourages $z$ and $s$ to have different information, i.e., the disentanglement of global and local factors. We call our method *CMI-maximizing regularization*, as it is the lower bound of the conditional mutual information (CMI) $I(x; z|s)$. Furthermore, we introduce an adversarial training technique for estimating the CMI. A simple way to estimate it would be considering $I(x; z)$ and $I(z; s)$ independently, but it might result in compounding approximation errors. Instead, we use the formularization of $\beta$-VAE and adversarial training (Ganin et al., 2016), which reduce the number of terms to be approximated. Specifically, we approximate the upper bound of $I(z; s)$ using a density ratio trick (DRT) (Nguyen et al., 2008), where an adversarial classifier models the density ratio. Once we estimate the bound, $I(z; s)$ can be minimized via backpropagation through the classifier.

In our experiments, we used DSAE and PixelCNN-VAE as illustrative examples of the SSM and ARM variants. In addition to evaluate the quality of global latent variable as in previous studies, we also evaluated the ability of controlled generation using a novel evaluation method inspired by Ravuri & Vinyals (2019). In the experiments, the CMI-maximizing regularization consistently outperformed the MI-maximizing one on image and speech datasets. These results support (i) our information theoretic view of learning global features: the sequential VAEs can suffer from obtaining redundant features when merely maximizing the MI. Also, the results support that (ii) regularizing $I(x; z)$ and $I(z; s)$ is complementary: learning global features can be facilitated by not only making $z$ informative, but also the control for which aspect of $x$ information (global or local) goes into $z$.

Our contribution can be summarized as follows: (i) through our analysis, we reveal the potential negative side-effect of MI-maximizing regularization, which has been standard in learning global representation with sequential VAEs. Then, the analysis encourages the sequential VAE community to seek for new regularization approach. (ii) In order to learn good global representation, we proposed regularizing $I(x; z)$ and $I(z; s)$ at the same time. $I(x; z)$ and $I(x; z)$ are robustly shown to work complementary by our experiments using two models and two domains (speech and image datasets). This finding would help improve various sequential VAEs proposed before.

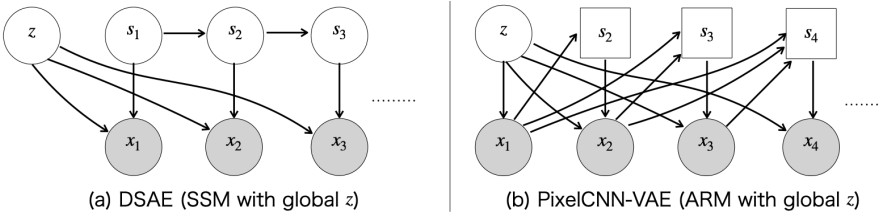

Figure 2: Graphical models for (a) DSAE and (b) PixelCNN-VAE.

## 2 PRELIMINARY

### 2.1 SEQUENTIAL VAES FOR LEARNING GLOBAL REPRESENTATIONS

Here we first explain the standard VAE; then, we give overviews of the DSAE and PixelCNN-VAE. Both models are shown to be interpreted as having two types of the latent variables, global $z$ and local $s_t$; although it is not explicitly stated for PixelCNN-VAE. Here, $s_t$ is designed to influence particular timesteps or dimensions of $x$ (e.g., a single-frame in a speech or a small area of pixels in an image). On the other hand, $z$ influences all the timesteps of $x$, although $z$ of DSAE and PixelCNN-VAE are imposed on different architectural constraints (discussed in Appendix C).

**Variational autoencoder (VAE)**   Let $p(x) := \int p(z)p(x|z)dz$ be a latent variable model, whose decoder $p(x|z)$ is parameterized by a deep neural network (DNN). Using an encoder distribution $q(z|x)$, which is also parameterized by a DNN, the VAEs maximize ELBO:

$$\mathcal{L}_{\text{ELBO}} := \mathbb{E}_{p_d(x)}\big[\mathbb{E}_{q(z|x)}[\log p(x|z)] - D_{\text{KL}}(q(z|x)||p(z))\big]. \tag{1}$$

Here, $p_d(x)$ denotes the data distribution. ELBO contains two terms: the reconstruction error and the Kullback-Leibler (KL) divergence between encoder $q(z|x)$ and the prior $p(z)$.

**Disentangled sequential autoencoder**   DSAE (Yingzhen & Mandt, 2018) is an extension of the SSMs for modeling the global and local features of sequential data as separate latent variables. Using the disentangled variables, DSAE can control the outputs (e.g., perform voice conversion). DSAE has a global latent variable $z$ and a local latent variable $s_t$, and generates an observation $x_t$ at time $t$ from $z$ and $s_t$. The ELBO can be expressed as follows:

$$\mathcal{L}_{\text{SSM}} := -\text{Recon} - \text{KL}(z) - \text{KL}(s),$$

$$\text{where} \quad \text{Recon} = -\mathbb{E}_{q(x,z,s)}\big[\sum_{t=1}^{T}\log p(x_t|s_t, z)\big], \quad \text{KL}(z) = \mathbb{E}_{q(x,z,s)}[D_{\text{KL}}(q(z|x_{\leq T})||p(z))],$$

$$\text{KL}(s) = \mathbb{E}_{q(x,z,s)}\big[\sum_{t=1}^{T}D_{\text{KL}}(q(s_t|x_{\leq T}, z, s_{t-1})||p(s_t|s_{t-1}))\big]. \tag{2}$$

Here, $p(s_t|s_{t-1})$ is a prior, $q(z|x_{\leq T})$ and $q(s_t|x_{\leq T}, z, s_{t-1})$ are encoders, $p(x_t|s_t, z)$ is a decoder, and $q(x, z, s) := p_d(x)q(z|x)q(s|x, z)$. Furthermore, $x_{<t}$ denotes all the elements of the sequences up to $t$, and $x$ denote $x := x_{\leq T}$. Figure 2-(a) illustrates the data generating process.

**VAE with PixelCNN decoder**   PixelCNN-VAE is designed to take advantage of both PixelCNN and VAEs. VAEs are known to fail in terms of capturing the local features of images, such as textures and sharp edges. Conversely, PixelCNN is good at capturing the local features, but often fails to generate globally coherent images and has no latent variables. Then, successfully trained PixelCNN-VAEs would generate high-fidelity data and induce latent variable $z$, which maintains only the global information by discarding the local information (Gulrajani et al., 2017).

PixelCNN-VAE can be interpreted as a structured VAE in which $x_t$ is generated from the global latent variable $z$ and local variable $s_t$ as follows. First, the autoregressive decoder is expressed as $p(x_{\leq T}|z) = \Pi_{t=1}^{T}p(x_t|z, x_{<t})$. This means that for every timestep $t$, $x_t$ is sampled from $p(x_t|z, x_{<t})$ using previous observations $x_{<t}$ and the latent variable $z$. Secondly, we assume that the decoder can be decomposed as $p(x_t|z, x_{<t}) = p(x_t|z, f(x_{<t}))$ using a neural network $f$ (PixelCNN). Finally, we introduce a random variable $s_t$ and its distribution $q(s_t|x_{<t}) = p(s_t|x_{<t}) :=$

$\delta(s_t - f(x_{<t}))$, where $\delta$ denotes the Dirac's delta, and $q(s_t|x_{<t}) = p(s_t|x_{<t})$ is employed to simplify the notation. With this notation, the decoder can be decomposed as $p(x_t|z, x_{<t}) = p(x_t|z, f(x_{<t})) = p(x_t|z, s_t)p(s_t|x_{<t})$ (details on the practical decomposition using PixelCNN have been provided in Appendix B). Thus, $x_t$ can be regarded to be generated from $z$ and $s_t$, which is sampled from $p(s_t|x_{<t})$ (see, Figure 2-(b)). Furthermore, the ELBO is given as follows:

$$\mathcal{L}_{\text{ARM}} = -\text{Recon} - \text{KL}(z). \tag{3}$$

## 2.2 Mutual information-maximizing regularization for Sequential VAEs

Despite the intentional data generating process of the sequential VAEs, the global latent variable $z$ often becomes uninformative. To alleviate this issue, MI-maximizing regularization methods are often employed to encourage $z$ to have $x$ information. Note that, here we consider the MI defined by the encoder (which corresponds to the representational MI in Alemi et al. (2018)):

$$I(x; z) = \mathbb{E}_{p_d(x)q(z|x)}\Big[\log \frac{p_d(x)q(z|x)}{p_d(x)q(z)}\Big], \tag{4}$$

A representative example of the MI-maximizing regularization is $\beta$-**VAE**, which was shown to work well in Alemi et al. (2018) and used as a baseline in He et al. (2019) (other methods are presented in Section 4). Because the ELBO (Eq. 1) contains a positive lower bound and a negative upper bound of $I(x; z)$, the MI can be controlled by balancing the two terms using a weighting parameter $\beta$. The concrete $\beta$-VAE objectives for DSAE and PixelCNN-VAE are:

$$\mathcal{V}_{\text{SSM}} := -\text{Recon} - \beta\text{KL}(z) - \text{KL}(s), \tag{5}$$
$$\mathcal{V}_{\text{ARM}} := -\text{Recon} - \beta\text{KL}(z). \tag{6}$$

Alemi et al. (2018) use $\beta < 1$ to regularize $I(x; z)$ to be large, although $\beta$-VAE was originally invented to encourage the independence of each dimension of $z$ with $\beta > 1$ by Higgins et al. (2017).

# 3 Proposed method

## 3.1 Decomposition of mutual information

Sequential VAEs with a global latent variable $z$ can in principle uncover global representation of data by exploiting its structured data generating process. Previous studies for the sequential VAEs have further regularized mutual information $I(x; z)$ to be large in order to alleviate posterior collapse (PC) (further discussed in Section 4). Unfortunately, the MI maximization is insufficient to uncover the global factor of variations, because it cannot control the type of information going into $z$. More specifically, as indicated in Section 1, the MI is decomposed as follows:

$$I(x; z) = I(x; z; s) + I(x; z|s) = I(z; s) - I(z; s|x) + I(x; z|s). \tag{7}$$

Simply maximizing $I(x; z)$ can increase $I(z; s)$ in the right-hand side, which is observed in our preliminary experiment in Appendix A. When $I(z; s)$ becomes large, $z$ is likely to have redundant local information, or conversely $s$ becomes to have global information. Which phenomenon occurs could depend on the network architecture, as discussed in Appendix C. In both cases, the performance of downstream tasks, e.g., classification from $z$ to the labels, or controlling the global characteristics of the decoder output using $z$, is likely to be degraded. Furthermore, as well as the MI $I(x; z)$, this MI $I(z; s)$ is defined by the encoder distribution $q(z, s)$ (Appendix E). Then, although the graphical model of DSAE is designed such that $z$ and $s$ are independent, $I(z; s)$ is not necessarily zero, i.e., $p(z, s) = p(z)p(s)$ does not necessarily mean $q(z, s) = q(z)q(s)$.

## 3.2 Conditional mutual information-maximizing regularization

Considering the limitations of MI regularization, we need a method that can encourage both (i) the increasing of $I(x; z)$ to prevent $z$ from becoming uninformative, and (ii) the decreasing of $I(z; s)$ to prevent $z$ and $s$ from having information that is irrelevant to the global and local structure, respectively. Therefore, we propose maximizing the following objective as a regularization approach:

$$I(x; z) - \alpha I(z; s). \tag{8}$$

As $I(z;s)$ measures the mutual dependence between $s$ and $z$, minimizing $I(z;s)$ encourages $z$ and $s$ not to have redundant information. Then, the induced global variable $z$ would have more $x$ information, while $z$ and $s$ maintains only the global and local information, respectively. The $\alpha$ is a weighting parameter for balancing the two terms. In practice, we found that $\alpha = 1$ works reasonably; therefore, we used $\alpha = 1$ for the reminder of the study.

Furthermore, it is noteworthy that our method is closely related to CMI estimation. Specifically, when assuming $\alpha \geq 1$,

$$I(x;z|s) = I(x;z) - I(z;s) + I(z;s|x) \geq I(x;z) - \alpha I(z;s) =: I_{\mathrm{CMI'}}. \tag{9}$$

It indicates that $I_{\mathrm{CMI'}} = I(x;z) - \alpha I(z;s)$ is equal to the lower bound of CMI (further discussed in Appendix D). CMI is known to be useful in selecting the features that are both individually informative and two-by-two weakly dependant (Fleuret, 2004). Therefore, maximizing $I(x;z|s)$ as regularization would make the features $z$ and $s$ informative but disentangled.

Then, we present one of the tractable instances to estimate $I_{\mathrm{CMI'}}$. A simple way to estimate $I_{\mathrm{CMI'}}$ may be to consider $I(x;z)$ and $I(z;s)$ independently; however, it must approximate both, $I(x;z)$ and $I(z;s)$, which may complicate optimization. Fortunately, when we assume $\alpha = 1$, we can reduce the number of terms to be approximated to only one, utilizing the $\beta$-VAE formularization. First, we express $I_{\mathrm{CMI'}}$ as follows (the derivation is given in Appendix F):

$$I_{\mathrm{CMI'}} = I(x;z) - I(z;s) = \mathbb{E}_{p_d(x)}\big[D_{\mathrm{KL}}(q(z|x)||p(z))\big] - D_{\mathrm{KL}}(q(z,s)||p(z)q(s)). \tag{10}$$

The first term is the same as KL($z$) in Eqs. 5 and 6, and is used for the $\beta$-VAE formularization. The second term is the upper bound of $I(z;s)$ because $D_{\mathrm{KL}}(q(z,s)||p(z)q(s)) = I(z;s) + D_{\mathrm{KL}}(q(z)||p(z))$.

Because the second term is difficult to calculate analytically, we estimate it using the DRT (Nguyen et al., 2008; Sugiyama et al., 2012), as performed in standard generative adversarial networks (Mohamed & Lakshminarayanan, 2017). By introducing the labels $y = 1$ for samples from $q(z,s)$ and $y = 0$ for those from $p(z)q(s)$, we re-express these distributions in conditional form, i.e., $q(z,s) := p(z,s|y=1)$ and $p(z)q(s) := p(z,s|y=0)$. The density ratio betwwen $q(z,s)$ and $p(z)q(s)$ can be computed using these conditional distributions as follows:

$$\frac{q(z,s)}{p(z)q(s)} = \frac{p(z,s|y=1)}{p(z,s|y=0)} = \frac{p(y=1|z,s)}{p(y=0|z,s)}, \tag{11}$$

where we used Bayes' rule and assumed that the marginal class probabilities are equal, i.e. $p(y=0) = p(y=1)$. Here, $p(y|z,s)$ can be approximated with a discriminator $D(z,s)$, which outputs $D = 1$ when $z, s \sim_{i.i.d.} q(z,s)$, and $D = 0$ when $z, s \sim_{i.i.d.} q(s)p(z)$. Then, Eq. 10 can be approximated as follows:

$$I_{\mathrm{CMI'}} \approx \mathbb{E}_{p_d(x)}[D_{\mathrm{KL}}(q(z|x)||p(z))] - \mathbb{E}_{q(z,s)}\Big[\log\frac{D(z,s)}{1-D(z,s)}\Big] =: I_{\mathrm{CMI-DRT}}. \tag{12}$$

We parameterize $D(z,s)$ with a DNN and train it alternately with the VAE objectives. Specifically, we train $D$ to maximize the following objective with Monte Carlo estimates:

$$\mathbb{E}_{q(z,s)}[\log D(z,s)] + \mathbb{E}_{p(z)q(s)}[\log(1 - D(z,s))].$$

Now, we can introduce the concrete objectives of the DSAEs and PixelCNN-VAEs with a CMI regularization term. Adding $I_{\mathrm{CMI-DRT}}$ as a regularization term to Eqs. 2 and 3 with a weighting parameter $\gamma$, we obtain the objective functions of our proposed method that need to be maximized:

$$\mathcal{J}_{\mathrm{SSM}} := \mathcal{L}_{\mathrm{SSM}} + \gamma I_{\mathrm{CMI-DRT}} = -\mathrm{Recon} - \mathrm{KL}(s) - (1-\gamma)\mathrm{KL}(z) - \gamma I'(z;s), \tag{13}$$

$$\mathcal{J}_{\mathrm{ARM}} := \mathcal{L}_{\mathrm{ARM}} + \gamma I_{\mathrm{CMI-DRT}} = -\mathrm{Recon} - (1-\gamma)\mathrm{KL}(z) - \gamma I'(z;s), \tag{14}$$

$$\text{where } I'(z;s) = \mathbb{E}_{q(z,s)}[\log\frac{D(z,s)}{1-D(z,s)}].$$

Considering $(1-\gamma)\mathrm{KL}(z)$ is equivalent to the weighting technique in $\beta$-VAE (note that $1-\gamma = \beta$, and see Eqs. 5 and 6), the proposed method consists of the $\beta$-VAE objective and $-I'(z;s)$. As noted in Section 2.2, $\beta$-VAE is effective for alleviating PC. However, because $\beta$-VAE alone is insufficient for decreasing the redundancy of $z$ and $s$, minimizing $I'(z,s)$ is employed.

Finally, we discuss alternative choices to estimate the second term of Eq. 10. While we chose to approximate the term with a discriminator, it can also be approximated with other distance such as maximum mean discripancy (MMD), or it can be minimized via Stein variational gradient (see, Zhao et al. (2019)). However, a weakness of these methods is that they are difficult to apply efficiently in high dimensions. Unfortunately, because the second term treats the random variable $[z, s_1, ..., s_T]$, the dimension size becomes high when $T$ is large. On the other hand, adversarial training requires only one assumption, i.e., $D$ perfectly approximates the true density ratio. In practice, while this assumption does not always hold true (Moyer et al., 2018; Iwasawa et al., 2020), it is also empirically known that original objectives (in our case, minimizing $D_{\mathrm{KL}}(q(z, s)||p(z)q(s))$) can be reasonably achieved (Ganin et al., 2016). In addition, various studies (e.g., Miyato et al. (2018); Iwasawa et al. (2020)) have proposed techniques to improve the robustness of adversarial training, and it has been shown to scale to high dimensions when carefully designed (Brock et al., 2019).

# 4 RELATED WORKS

This study is closely related to the literature on disentangled representation. Locatello et al. (2019) claimed that pure unsupervised disentangling (Chen et al., 2016; Higgins et al., 2017; Kim & Mnih, 2018) is fundamentally impossible, whereas using rich supervision (Kulkarni et al., 2015) can be costly. Thus, the use of inductive bias or weak supervision (Shu et al., 2020) has been encouraged. The assumption that data are generated from global and local factors is a representative example of the inductive bias. Such data generating process can be probabilistically expressed by the sequential VAEs with a global latent variable. Then, the sequential VAEs have been studied for disentangling styles and topics of texts (Bowman et al., 2016), object identities from the detailed textures of images (Chen et al., 2017), content and motion of movies (Hsieh et al., 2018), and the speaker and linguistic information of speeches (Hsu et al., 2017; Yingzhen & Mandt, 2018). Although this paper focused on DSAE and PixelCNN-VAE as examples, our method could be also combined with them.

Unfortunately, the design of the structured data generating processes alone is often insufficient to learn the global features. To address this issue, Bowman et al. (2016); Chen et al. (2017) initially proposed to weaken the decoder because PC often occurs when using highly expressive decoders. Subsequently, various methods have been proposed to control the MI $I(x; z)$ with a regularization term, which does not require problem-specific architectural constraints of Bowman et al. (2016); Chen et al. (2017). Concrete examples of MI-maximizing regularization methods are as follows: **InfoVAE**: (Zhao et al., 2019) estimates $I(x; z)$ using the MMD or adversarial training. $\beta$-**VAE**: Alemi et al. (2018) proposed targeting a specific rate (the KL term value) via $\beta$-VAE , and observed that the objective with $\beta < 1$ produces solutions to alleviate PC. $\beta$-VAE is a simpler than InfoVAE since it does not require an approximation of $I(x; z)$. **Auxiliary loss**: (Lucas & Verbeek, 2018) uses the auxiliary tasks of predicting $x$ from $z$, which approximates the minimization of conditional entropy $H(x|z)$. The minimization of $H(x|z)$ is equivalent to maximizing $I(x; z)$ because the data entropy $H(x)$ is constant. **Discriminative objective**: (Hsu et al., 2017) predicts a sequence index from $z$, which also approximates $H(x|z)$ minimization in the finite sample case.

Various studies have attemted to separate relevant from irrelevant information via information-theoretic regularization. Namely, the studies in the literature regarding domain-invariant representation proposed to learn the invariant representation using adversarial training (Ganin et al., 2016; Xie et al., 2017; Liu et al., 2018), variational information bottleneck frameworks (Moyer et al., 2018), or Hilbert-Schmidt independence criterion (Jaiswal et al., 2019). Our regularization term of minimizing $I(z; s)$ is inspired and similar to these studies; however, it differs in considering PC at the same time (i.e., maximizing $I(x; z)$). Also, the separation could be achieved by the design of network architectures, as was performed in VQ-VAE2 (Razavi et al., 2019). Our proposal is the regularization term and orthogonal to such architecture choices.

Also, our analysis is similar to that of Moyer et al. (2018), but differs in two ways. Firstly, while the phenomenon that "large $I(x; z)$ results in large $I(z; s)$" was discussed by them, the whole mechanism that "MI-maximizing regularization for alleviating PC has a negative side-effect to increase $I(z; s)$" has been overlooked in the sequential VAE community. Secondly, by explicitly considering the relationship between the two latent variables $z$ and $s$, our analysis is able to highlight a new problem. For example, Moyer et al. (2018) consider the relationship between the latent variable $z$ and the observed nuisance factor $s$. Then, their focus is only on removing the redundant information

from $z$. On the other hand, our analysis highlights the need to consider removing the redundant information from $s$ at the same time as removing the redundant information from $z$. Although the former has been overlooked, it is an important issue in applications such as voice conversion.

Some studies have also proposed methods for alleviating PC, which are complementary to MI maximization. Lucas et al. (2019) argued that the variance of the decoder influences the stability of local stationary points corresponding to PC. He et al. (2019) proposed a method that remedies ill-training-dynamics. Our study differs in aiming at obtaining informative and *disentangled* representation with sequential VAEs, although they could, in principle, be combined with our method.

From technical perspective, our work is also related to a feature selection technique based on CMI (Fleuret, 2004). CMI is known to be useful in selecting the features that are both individually informative and two-by-two weakly dependant. Then, the CMI-based technique is different from the MI-based one in considering the independence of the features. Moreover, it is different from previous studies for disentangled representation learning, e.g., Higgins et al. (2017); Kim & Mnih (2018); Liu et al. (2018) control only the independence of latent factors. Also, Mukherjee et al. (2019) first proposed the estimation of CMI using DNNs; however, our method is different in utilizing the encoder distribution of VAEs to improve the estimation (Zhao et al., 2019; Poole et al., 2019).

## 5 EXPERIMENTS

### 5.1 SETTINGS

We performed experiments to confirm the effect of regularizing both $I(x; z)$ and $I(z; s)$ for learning good representation, using DSAE and PixelCNN-VAE as representative examples of the sequential VAEs. We used the speech corpus TIMIT (Garofolo et al., 1992) for the DSAE, and evaluated representation quality using a speaker verification task, as was performed in previous studies (Hsu et al., 2017; Yingzhen & Mandt, 2018). For PixelCNN-VAE, we trained the VAE with a 13-layer Pixel-CNN decoder on the statically binarized MNIST and Fashion-MNIST (Xiao et al., 2017) datasets. Using the trained models, we performed linear classification from $z$ to class labels to evaluated representation quality, as was performed in (Razavi et al., 2019), and then evaluated the ability of controlled generation. $z$, which has a dimensional size of 32, was concatenated with the feature map outputted from the fifth layer of the PixelCNN (which corresponds to $s$, see Appendix B), and was passed to the sixth layer. Further details are given in Appendix G.

As the proposed method, we employed the objective functions $\mathcal{J}_{\text{SSM}}$ and $\mathcal{J}_{\text{ARM}}$ in Eqs. 13 and 14 (denoted as **CMI-VAE**). We implemented a discriminator $D$ as a CNN that receives $s$ and $z$ as inputs (Appendix I), and trained it alternately with the VAEs. As baseline methods, we employed $\beta$**-VAE** (see, Section 2.2). The objectives of $\beta$-VAE are given in Eqs. 5 and 6, and are equal to CMI-VAE, except for not having the $I'(z; s)$ term. Moreover, we employed the regularization method proposed in Makhzani & Frey (2017); Zhao et al. (2019), which directly estimates and maximizes $I(x; z)$ with adversarial training (denoted as **MI-VAE**). In the method, $I(x; z)$ is added to ELBO (Eqs. 2 and 3) as a regularization term, along with a weighting term $\gamma$ (details can be found in Appendix H). The performances of the models were verified by changing the value of $\gamma$.

### 5.2 SPEAKER VERIFICATION WITH DISENTANGLED SEQUENTIAL AUTOENCODERS

For a quantitative assessment of the global representation of DSAE, we evaluate whether $z$ can uncover the speaker individualities, which are the global features of speech. Specifically, we extract $z$ and $s_{\leq T}$ from the test utterances using the mean of the encoders of the learned DSAE. Subsequently, we performed speaker verification by measuring the cosine similarity of the variables and evaluated the equal error rate (EER). Here, EER is measured for both $z$ and $s$ (denoted as EER($z$) and EER($s$), respectively), and $s_{\leq T}$ is averaged over each utterance prior to its measurement. A lower EER($z$) is preferable because it indicates that the model has an improved global representation, containing sufficient information of the speakers in a linear-separable form. Furthermore, a higher EER($s$) is preferable because it indicates that $s$ does not have the redundant speaker information. In addition, we report KL($z$) (see, Eq. 2), which approximates the amount of information in $z$.

Table 1 presents the values of KL($z$) and EER for the vanilla DSAE, $\beta$-VAE, and CMI-VAE. Note that our results for vanilla DSAE differ from those reported in Yingzhen & Mandt (2018) (DSAE*

Table 1: KL term and EER values of DSAE trained using TIMIT. Each model was trained with a weight $\gamma$. The $\uparrow$ and $\downarrow$ indicate that the purpose was to obtain a high and low score, respectively.

| Model | $\gamma$ | KL($z$) | EER($z$) $\downarrow$ | EER(s) $\uparrow$ |
|---|---|---|---|---|
| DSAE* | 0.00 | - | 4.82 | 18.89 |
| DSAE(our implementation) | | 18.00 | 11.01 | 18.64 |
| + $\beta$-VAE | 0.40 ($\beta = 0.6$) | 53.28 | 3.88 | 29.45 |
| + CMI-VAE | | 54.13 | **3.43** | **30.96** |
| + $\beta$-VAE | 0.80 ($\beta = 0.2$) | 145.88 | 4.33 | 38.84 |
| + CMI-VAE | | 145.09 | **3.99** | **41.30** |
| + $\beta$-VAE | 0.90 ($\beta = 1e-1$) | 202.52 | 4.55 | 39.42 |
| + CMI-VAE | | 199.89 | **4.39** | **41.25** |
| + $\beta$-VAE | 0.99 ($\beta = 1e-2$) | 364.71 | 6.33 | 38.63 |
| + CMI-VAE | | 361.03 | **5.06** | **40.08** |

in the table), which may be due to differences in the unreported training settings. The table presents that (1) a lower $\gamma$ (such as 0) provides a lower EER(s), which indicates that $s$ have global information instead of $z$ owing to PC, without regularizing $I(x; z)$. Furthermore, (2) given a fixed $\gamma$, CMI-VAE consistently achieved a lower EER($z$) and a higher EER(s) while having the same level of KL($z$) compared to $\beta$-VAE. This indicates that regularizing $I(z; s)$ is complementary to MI-maximization ($\beta$-VAE), yielding a better $z$ and $s$ that have sufficient global or local information but are well compressed. Note that $\gamma \geq 0.8$ yields a higher EER($z$) than $\gamma = 0.4$, which may be due to the fact that the independence of each dimension of $z$ is worsened by increasing $\gamma$, as indicated in Higgins et al. (2017), and the induced non-linear relation cannot be measured by the cosine similarity. In fact, $\gamma \geq 0.8$ presented a better performance in the voice conversion experiment in Appendix J, indicating that $z$ with $\gamma \geq 0.8$ has more global information, although the EER($z$) is lower. Also, note that the EER($Z$) reported in Hsu et al. (2017) is lower than the results for CMI-CAE here. However, we believe that our claim, "regularizing $I(x; z)$ and $I(z; s)$ is complementary", is defended even if we could not achieve state-of-the-art results.

## 5.3 VAEs with PixelCNN decoder

**Unsupervised learning for image classification** For a quantitative assessment of the representation $z$ of PixelCNN-VAEs, we performed a logistic regression from $z$ to the class labels $y$ on MNIST and Fashion-MNIST. Specifically, first, we extracted $z$ from 1000 training samples using the mean of $q(z|x)$, where each of the 10 classes had 100 samples, and trained the classifier with a total of 1000 samples. Then, we evaluated the acccuracy of the logistic regression (AoLR) on the test data. A high AoLR indicates that $z$ succeeds in capturing the label information in a linear-separable form.

Figures 3(a) and 3(b) present AoLR for $\beta$-VAE, MI-VAE, and CMI-VAE, along with the ELBO and KL($z$). In the figures, an upper left curve indicates that the method balance better compression (low KL($z$)) and high downstream task performance. As shown in the figures, given a fixed $\gamma$, the AoLRs for CMI-VAE are consistently better than those for $\beta$-VAE and MI-VAE, although all the methods have the same level of KL($z$). This indicates that CMI-VAE can extract more *global* information when compressing data to the same size as $\beta$-VAE does. Note that a small $\gamma$ (such as $\gamma = 0$) and very large $\gamma$ degrade the AoLRs, which may be attributed to the same reason as explained in Section 5.2. Furthermore, the AoLRs of MI-VAE are lower than those of $\beta$-VAE, which may be due to the adversarial training in MI-VAE causing optimization difficulties, as stated in Alemi et al. (2018).

**Controlled generation** Most previous works (Yingzhen & Mandt, 2018; He et al., 2019) have primarily focused on evaluating the quality of global representation. However, a better representation does not necessarily improve the performance of the controlled generation, as Nie et al. (2020) claimed. Then, to evaluate the ability of the controlled generation, we propose a modified version of the classification accuracy score (CAS) (Ravuri & Vinyals, 2019), called mCAS. CAS trains a classifier which predicts class labels only from the samples generated from conditional generative models, and then evaluates the classification accuracy on real images, thus measuring the sample quality and diversity of the model. CAS is not directly applicable to non-conditional models such as PixelCNN-VAEs. Instead, mCAS measures the ability of the model to produce high quality, diverse, but globally coherent (i.e., belonging to the same class) images for a given $z$.

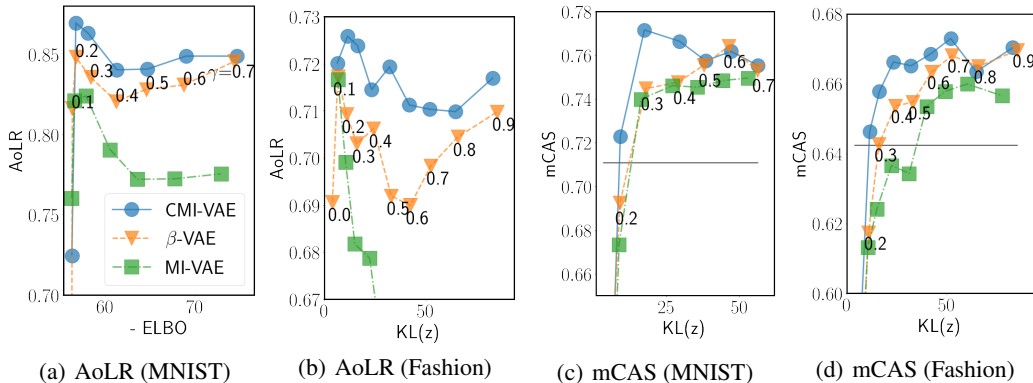

(a) AoLR (MNIST)    (b) AoLR (Fashion)    (c) mCAS (MNIST)    (d) mCAS (Fashion)

Figure 3: Comparison of CMI-VAE with $\beta$-VAE and MI-VAE. Each maker for $\beta$-VAE is annotated with the value of $\gamma$. In the figures, an upper left curve is desirable because it shows the method balance better compression (low KL($z$)) and high downstream task performance (AoLR and mCAS, see explanations in Section 5.3). Also, detailed results can be found in Appendix K.2.

In mCAS, we first prepared 100 real images $\{x_i\}_{i=1}^{100}$, along with their class labels $\{y_i\}_{i=1}^{100}$, where each of the 10 classes had 10 samples. Then, using the trained VAEs, we encoded each $x_i$ into $z_i$, and decoded $z_i$ to obtain 10 images $\{\hat{x}_{i,j}\}_{j=1}^{10}$ for every $z_i$, thereby resulting in 1000 generated images (sample images $\hat{x}$ can be found in Appendix K.3). Finally, we trained the logistic classifier with the pairs $\{(\hat{x}_{i,j}, y_i)|i \in \{1, ..., 100\}, j \in \{1, ..., 10\}\}$ and evaluated the performance on real test images. Intuitively, when the decoder ignores $z$, the generated samples might belong to a class different from the original ones, which produces label errors. Moreover, when $z$ has excessive information regarding $x$ and the VAE resembles an identity mapping, the diversity of the generated samples decreases (recall that 10 samples are generated for every $z_i$), which induces overfitting of the classifier. Therefore, to achieve a high mCAS, $z$ should capture only the global (label) information.

Figure 3(c) and 3(d) compares the mCAS along with KL($z$) on MNIST and Fashion-MNIST. In addition, the black horizontal line indicates the classification accuracy when the classifier is trained on 100 real samples $\{(x_i, y_i)\}_{i=1}^{100}$, and evaluated on real test images, which will be referred to as the *baseline score*. The following can be observed from the figures: (1) The mCAS of the three methods outperformed the baseline score, despite using only 100 labeled samples, as well as in the baseline score, indicating that properly regularized PixelCNN-VAEs could be used for data augmentation. (2) As expected, a significantly low KL($z$) gives a low mCAS because the decoder of the VAE does not utilize $z$. Moreover, a significantly high KL($z$) also tends to degrade mCAS, because the decoder might resemble a one-to-one mapping from $z$ to $x$ and therefore, degrade the diversity. This phoenomenon can also be observed in the sample images in appendix K.3: there seems to be little diversity in samples drawn from $\beta$-VAE and CMI-VAE with $\gamma = 0.6$. (3) The curves for CMI-VAE are consistently left to those for $\beta$-VAE, indicating that regularizing $I(z; s)$ is also complementary to regularizing $I(x; z)$ at the controlled generation.

## 6 DISCUSSIONS AND FUTURE WORKS

Based on the experimental results, it was confirmed that regularizing $I(z; s)$ is complementary to regularizing $I(x; z)$, and leads to an improvement in the learning of global latent variables. Here, we chose to extend $\beta$-VAE to construct the proposed objective function because we believe $\beta$-VAE is the simplest MI-maximization method that requires fewer hyperparameters, widely used in (sequential) VAE community (e.g., He et al. (2019); Alemi et al. (2018)). However, other MI estimation methods, such as discriminative objective and MMD-based InfoVAE, can be extended to CMI regularization by the addition of the $I(z; s)$ minimization term (see, Section 3.2). Incorporating such MI maximization methods into the estimation of CMI, or stabilizing adversarial training with some technique (Miyato et al., 2018) might improve the performance, and this remains an issue to be addressed in a future work. Also, it would be interesting to approximate $I(x; z)$ and $I(z; s)$ separately, and tune the strength of them independently. Future studies may also apply the proposed method to encourage the learning of the representation that captures the global factors of the environment such as maps, to support reinforcement learning, as suggested in Gregor et al. (2019).

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

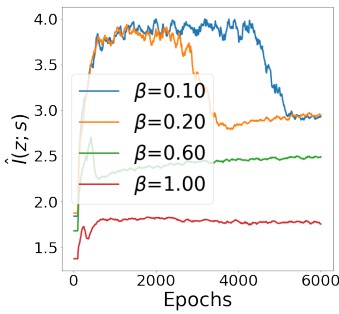

Figure 4: $\hat{I}(z; s)$ values of DSAE trained with $\beta$-VAE objective.

# APPENDIX

## A  EMPIRICAL EVIDENCE OF LIMITATION OF MI-MAXIMIZING REGULARIZATION

In this section, we evaluate the $I(z; s)$ values of DSAE, and the MI-maximizing regularization is shown to result in increasing it, contrary to the intention of learning the disentangled global features. Note that the $I(z; s)$ of DSAE ideally equals to zero because its graphical model is designed such that $z$ and $s$ are independent. However, $I(z; s)$ is not necessarily zero because the representational MI is considered (see, also, Section 3.1). Because $I(z; s)$ is intractable, we used the value of $\hat{I}(z; s)$ estimated with DRT was used in a similar manner to Section 3.2. Namely, we used the following equation:

$$\frac{q(z, s)}{q(z)q(s)} =: \frac{p(z, s|y = 1)}{p(z, s|y = 0)} = \frac{p(y = 1|z, s)}{p(y = 0|z, s)}, \text{ where } p(y = 0) = p(y = 1) = \frac{1}{2}. \quad (15)$$

$p(y|z, s)$ can be approximated with a discriminator $D(z, s)$ that outputs $D = 1$ when $z, s \sim_{i.i.d.} q(z, s)$, and $D = 0$ when $z, s \sim_{i.i.d.} q(s)q(z)$. Then, $I(z; s)$ can be approximated as follows:

$$I(z; s) \approx \mathbb{E}_{p_d(x)q(z,s|x)}\left[\log \frac{D(z, s)}{1 - D(z, s)}\right] =: \hat{I}(z; s). \quad (16)$$

$D(z, s)$ is parameterized with a DNN, and trained alternately with the VAEs' objectives. The other training settings are the same as those in Section 5.2, and can be found in Appendix G.

Figure 4 presents $\hat{I}(z; s)$ values of DSAE trained with the $\beta$-VAE objective (Eqs. 5 and 6), where KL($z$) is reweighted with parameter $\beta = 1 - \gamma$. The figure indicates that if $\beta \geq 0.2$, a smaller $\beta$ results in a larger $\hat{I}(z; s)$. This indicates that when we simply regularize $I(x; z)$ to be large using $\beta$-VAE objective, $z$ and $s$ become to have redundant information. In contrast, when we do not regularize $I(x; z)$ (i.e., $\beta = 1$), $\hat{I}(z; s)$ becomes small; however, $z$ also becomes uninformative regarding $x$.

One may notice that even if $\beta$ is smaller than 0.2, $\hat{I}(z; s)$ will not increase. This is probably due to the following reason. Firstly, in DSAE, it is difficult for the $z$ to have redundant local information due to the architectural constraint (Appendix C). Then, the $\hat{I}(z; s)$ might become larger when $s$, not $z$, has redundant information. That is, regardless of the value of $\beta$, $s$ may retain a certain degree of global information due to PC (which is supported by the experiments in Section 5.2). Therefore, when $z$ has global information as $\beta$ decreases, $I(z; s)$ increases. However, once $z$ has enough global information, there is no room for $I(z; s)$ to increase beyond a certain point.

## B  PRACTICAL DECOMPOSITION OF AUTOREGRESSIVE DATA GENERATING PROCESS

Here we show the practical decomposition of the autoregressive data generating process $\Pi_{t=1}^{T} p(x_t|z, x_{<t}) = \Pi_{t=1}^{T} p(x_t|z, s_t)q(s_t|x_{<t})$ for PixelCNN. We consider the 13-layer PixelCNN

used in He et al. (2019), which has five (7 x 7)-kernel-size layers, followed by, four (5 x 5) layers, and then four (3 x 3) layers. Each layer has 64 feature maps with dimensions 28 x 28 dimensions. The latent variable $z$ is extracted by an encoder, linearly transformed into (28, 28, 4) feature maps, and then concatenated to the each layer of the PixelCNN feature maps after the sixth layer. We denote the output of the $i$-th ($i \in \{1, ..., 13\}$) layer as $h_{i,t}$, where $t$ denotes the timestep (x and y coordinates, and $t \in \{1, ..., 28 \times 28 = 784\}$). Then, we can put $s_t := h_{6,<t}$ and the decomposition $\Pi_{t=1}^{T} p(x_t|z, x_{<t}) = \Pi_{t=1}^{T} p(x_t|z, s_t) q(s_t|x_{<t})$ holds because only $h_{6,<t}$ (not $h_{6,\geq t}$) are used to generate $x_t$ with causal convolution.

One might wonder whether the activations of the PixelCNN, which is the deterministic function of $x$, can be treated as random variables. However, because we regularize $s$ only via minimizing $I(z; s)$ (Section 3.2), $s$ can be determined to be treated as random variables. specifically, $I(z; s)$ is defined by the joint distribution $q(z; s) = \int p_d(x) q(z|x) q(s|x, z) dx$ (see definition in Appendix E), where $q(s|x, z) = q(s|x) = \Pi_{t=1}^{T} \delta(s_t - f(x_{<t}))$ would be integrated over a random variable $x$. Therefore, $z$ and $s$ have no deterministic relation and $s$ can be meaningfully referred to as local latent variables. Furthermore, it is common to treat the activations of hidden layers as random variables and to consider their MI (or conditional entropy) in the literature on domain-invariant representation learning (Xie et al., 2017).

Note that, the definition of $s$ is an important factor for the "control" of what will be learned in $z$; however, anything is acceptable as long as $s_t$ has sufficiently large receptive fields. For example, Chen et al. (2017) proposed improving global representation $z$ by using smaller receptive fields for $q(s_t|x_{<t})$ and constraining $s_t$ to more local information. Although this architectural constraint can make $z$ informative, it requires weakening the expressiveness of PixelCNN and can degrade ELBO (Chen et al., 2017). By contrast, our method can be applied regardless of the size of the receptive fields because it prevents $s$ from having global information with an information theoretic regularization term. Therefore, the architectural change of Chen et al. (2017) was not employed and large receptive fields were used to balance sufficient ELBO and representation quality.

## C  DSAE AND PIXELCNN-VAE HAVE DIFFERENT ARCHITECTURAL CONSTRAINTS ON GLOBAL LATENT VARIABLES

$z$ of DSAE and PixelCNN-VAE are imposed on different architectural constraints. In DSAE, the $z$ is constrained to have no local information. On the other hand, the $z$ of PixelCNN-VAEs has no such architectural constraints, although it is designed to capture global features via the structured data generating process. Here, we distinguish *the structured data generating process* from *the architectural constraints*: the former is the constraint based on the probabilistic graphical model, while the latter is the constraint based on the neural network structures. Specifically, the $z$ of DSAE is concatenated with $s_t$ and fed into the fully connected neural network decoder for all timestep $t \in \{1, ..., T\}$, so the $z$ may have the same effects on each timestep $t$. On the other hand, the $z$ of PixelCNN-VAE is linearly transformed into (28, 28, 4) feature maps, and then concatenated to the each layer of PixelCNN feature maps (see, Appendix B). Since the linear transformation creates the feature maps that depend on timesteps (x and y coordinates), it becomes easy for the $z$ to have different effects on each timestep $t$. Note that, such linear transformation is commonly employed in previous studies of PixelCNN-VAEs (e.g., He et al. (2019)) in order to improve expressiveness of the decoder.

Due to these architectural differences, different phenomena can be observed in DSAE and PixelCNN-VAE when using MI-maximizing regularization. First, both the models have in common that $I(z; s)$ would become larger when using the MI regularization (see, Section 3.1). However, in DSAE, it is likely that $s$ has redundant global features, not that $z$ has redundant local features, because it is difficult for the $z$ to have local information due to the architectural constraint. On the other hand, in PixelCNN-VAE, $z$ can have redundant local features. Then, these architectural differences would cause different problems in the controlled generation using DSAEs and PixelCNN-VAEs. DSAE can hopefully change speaker individualities while preserving the linguistic contents (i.e., perform voice conversion), by swapping the $z$ of two utterances and reconstructing them. However, if $s$ still contains speaker information due to the redundancy, the decoder can extract speaker information from either $s$ or $z$ and there is no guarantee that $z$ will be used (see, also, Appendix J). For PixelCNN-VAE, previous studies (Alemi et al., 2018; Razavi et al., 2019) have shown that by

stochastically sampling $x$ from PixelCNN-VAE with a given $z$, one can obtain images with different local patterns but similar global characteristics (e.g. color background, scale, and structure of objects). However, when $z$ has all (local and global) information, the diversity of the generated images would decrease, because the decoder resembles one-to-one mapping from $z$ to $x$ (see, also, Section 5.3).

## D   THE LOWER BOUND OF $I(x; z|s)$

Here we derive Eq. 9 and discuss the approximation error between $I(x; z|s)$ and $I_{\mathrm{CMI'}}$. Firstly, we can take the lower bound:

$$I(x; z|s) = I(x; z) - I(z; s) + I(z; s|x) \geq I(x; z) - I(z; s), \tag{17}$$

since the MI $I(z; s|x)$ is positive. Then, the lower bound $I(x; z) - I(z; s)$ has approximation error $I(z; s|x)$. Note that the error can be small under a particular condition. Namely, the error can be decomposed as:

$$I(z; s|x) = H(z|x) - H(z|x, s).$$

Here, both $H(z|x)$ and $H(z|x, s)$ is thought to be small when $x$ is high-dimensional data such as images, movies, and audios, because observing such $x$ would enable us to predict $z$ accurately. Also, empirically, it has been shown that the performance of inference model did not drop much even if the encoders of DSAE are decomposed into $q(z, s|x) = q(z|x)q(s|x)$ (Yingzhen & Mandt, 2018), which indicates the error is small.

In addition, as long as $\alpha \geq 1$, the following condition holds:

$$I(x; z) - I(z; s) \geq I(x; z) - \alpha I(z; s), \tag{18}$$

because the MI $I(z; s)$ is positive. This approximation error becomes the smallest when $\alpha = 1$.

## E   DEFINITION OF $I(z; s)$

This paper considers the MI $I(z; s)$ defined by the encoder (which corresponds to representational MI in Alemi et al. (2018)). Namely,

$$I(z; s) = \mathbb{E}_{q(z,s)}[\log \frac{q(z)q(s|z)}{q(z)q(s)}], \tag{19}$$

where the joint distribution is $q(z; s) \coloneqq \int p_d(x)q(z|x)q(s|x, z)dx$.

## F   DERIVING EQ. 10

Here we present the deriviation of Eq. 10:

$$I(x; z) - I(z; s) = \mathbb{E}_{q(x,z)}[\log \frac{q(z|x)}{q(z)}] - \mathbb{E}_{q(z,s)}[\log \frac{q(z|s)}{q(z)}] \tag{20}$$

$$= \mathbb{E}_{q(x,z,s)}[\log \frac{q(z|x)q(z)p(z)}{q(z|s)q(z)p(z)}] \tag{21}$$

$$= \mathbb{E}_{q(x,z)}[\log \frac{q(z|x)}{p(z)}] - \mathbb{E}_{q(z,s)}[\log \frac{q(z, s)}{p(z)q(s)}] \tag{22}$$

$$= \mathbb{E}_{p_d(x)}[D_{\mathrm{KL}}(q(z|x)||p(z))] - D_{\mathrm{KL}}(q(z, s)||p(z)q(s)). \tag{23}$$

## G   DETAILS OF EXPERIMENTAL SETTINGS

### G.1   DISENTANGLED SEQUENTIAL AUTOENCODER

**Data preprocessing**   We use the TIMIT data (Garofolo et al., 1992), which contains broadband 16kHz recordings of phonetically-balanced read speech. A total of 6300 utterances (5.4 hours) are

presented with 10 sentences from each of the 630 speakers (70% male and 30% female). Garofolo et al. (1992) have originally split the data into train/test subset, and we further split the train subset into 90% of train and 10% of validation subset. We followed Hsu et al. (2017); Yingzhen & Mandt (2018) for data preprocessing: the raw speech waveforms are first split into sub-sequences of 200ms, and then preprocessed with sparse fast Fourier transform to obtain a 201 dimensional log-magnitude spectrum, with the window size 25ms and shift size 10 ms. This results in $T = 20$ for the observation $x_{1:T}$.

**Optimization**  we follow Yingzhen & Mandt (2018) for model architecture, data preprocessing, and evaluation procedures. The dimensionality of $s_t$ and $z$ were fixed at 64; we set $T = 20$ for the observation $x_{\leq T}$. We used Adam optimizer with learning rate 2e-4 for the VAE and 2e-3 for the discriminator, and trained the models for 6000 epochs to get good convergence on the training set. The VAE architecture followed *full model* in Yingzhen & Mandt (2018), and the discriminator architecture is described in Appendix I. The discriminator is updated twice while the VAE is updated once. The results are averaged over three random seed trials.

### G.2  PIXELCNN-VAE

**Data preprocessing**  We use the statically binarized version of MNIST and Fashion-MNIST datasets: each pixel value $\in [0, 1]$ is binarized with the threshold 0.5. The datasets are originally split into train/test subsets, and we further split the train subsets into 80% of train and 20% of validation subsets.

**Optimization**  Regarding the optimization of VAEs, we used the Adam optimizer with a learning rate of 0.0001, trained for 300 epochs. We reported the values for the test data when the objective function for the validation data was maximized. Regarding the discriminators, we used the Adam optimizer with learning rate 0.001. The discriminator architecture is described in Appendix I, and is updated twice while the VAE is updated once. As for the PixelCNN architecture, see Appendix B.

## H  DETAILS OF MI-VAE IN OUR EXPERIMENTS

We employ $I(x; z)$ maximization method proposed by Makhzani & Frey (2017); Zhao et al. (2019) as a baseline method in our experiment. Briefly, we add $I(x; z)$ to the standard VAE objectives as a regularization term with weighting term $\gamma$.

To estimate $I(x; z)$, Makhzani & Frey (2017) utilize the follwing relation based on the density ratio trick:

$$\frac{q(z)}{p(z)} =: \frac{p(z|y=1)}{p(z|y=0)} = \frac{p(y=1|z)}{p(y=0|z)}, \tag{24}$$

where $p(y = 0) = \frac{1}{2}$ and $p(y = 1) = \frac{1}{2}$. Then, although conditional probability $p(y|z)$ cannot be obtained, it can be approximated with a discriminator $D(z)$, which outputs $D = 1$ when $z \sim_{i.i.d.} q(z)$ and $D = 0$ when $z \sim_{i.i.d.} p(z)$. Then, $I(x; z)$ can be approximated as follows:

$$I(x; z) = \mathbb{E}_{p_d(x)}[D_{KL}[q(z|x)||p(z)] - D_{KL}(q(z)||p(z))]$$

$$\approx \mathbb{E}[D_{KL}[q(z|x)||p(z)] - \log \frac{D(z)}{1 - D(z)}]$$

$$=: I_{\text{MI-DRT}}. \tag{25}$$

$D(z)$ is parameterized with some DNN, and trained alternately with VAEs' objectives. Namely, $D$ is trained to maximize the following objective with Monte Carlo sampling:

$$\mathbb{E}_{q(z)}[\log D(z)] + \mathbb{E}_{p(z)}[\log(1 - D(z))].$$

Finally, we introduce the concrete objective of PixelCNN-VAE with the regularization term $I_{MI}$. Adding $I_{\text{MI-DRT}}$ to the objective of PixelCNN-VAE, we obtain the objective functions of MI-VAE:

$$\mathcal{W}_{\text{ARM}} := \mathcal{L}_{\text{ARM}} + \gamma I_{\text{MI-DRT}} = -\text{Recon} - (1 - \gamma)\text{KL}(z) - \gamma \mathbb{E}[\log \frac{D(z)}{1 - D(z)}]. \tag{26}$$

In short, the objective only differs from our CMI maximization method in that the discriminator is added on the purpose of minimizing $D_{KL}(q(z)||p(z))$, while our method minimizes $D_{KL}(q(z,s)||p(z)q(s))$ and encourages disentanglement of $z$ and $s$.

## I  DISCRIMINATOR SETTINGS

We have used discriminators for CMI-VAE and MI-VAE (whose details can be found in Appendix H). For the discriminator of CMI-VAE, we first applied convolutional encoder and took mean pooling, obtaining the embedding of $s_{1:T}$. Then, in PixelCNN-VAE, we took innner product of the embedding and $z$ and treated it as logit of the discriminator. On the other hand, in DSAE, we took cosine similarity of the embedding and $z$, multiplied the similarity by a learnable scale parameter, and treated it as logit of the discriminator. The encoder architectures for PixelCNN-VAE and DSAE are summarized as follows, with the format Conv (depth, kernel size, stride, padding):

**PixelCNN-VAE**

- Input (28, 28, 1)
- Conv2D (256, 4, 2, 1)
- BatchNorm
- ReLU
- Conv2D (256, 4, 2, 1)
- BatchNorm
- ReLU
- Conv2D ($z$-dim, 4, 2, 1)

**DSAE**

- Input (20, 201)
- Conv1D (256, 4, 2, 1)
- BatchNorm
- ReLU
- Conv1D (256, 4, 2, 1)
- BatchNorm
- ReLU
- Conv1D ($z$-dim, 4, 2, 1)

The discriminator architecture for MI-VAE is summarized as follows, with the format Linear (input size, output size):

- Input ($z$-dim)
- Linear ($z$-dim, 400)
- ReLU
- Linear (400, 1)
- Softmax

## J  VOICE CONVERSION EXPERIMENTS USING DISENTANGLED SEQUENTIAL AUTOENCODER

For a quantitative assessment of controlled generation by DSAE, we performed voice conversion and evaluated the models with a score similar to mCAS (see, Section 5.3), which we call VC-mCAS. First, we prepared 500 real speeches (spectrograms with $T = 20$) $\{x_i\}_{i=1}^{500}$, along with their gender

Table 2: VC-mCAS for $\beta$-VAE and CMI-VAE.

| Model | $\gamma$ | VC-mCAS(mean) | VC-mCAS(max) |
|---|---|---|---|
| DSAE + $\beta$-VAE | 0.4000 | 83.73 | 85.8 |
| DSAE + CMI-VAE | 0.4000 | **84.33** | **86.6** |
| DSAE + $\beta$-VAE | 0.8000 | 87.27 | 87.6 |
| DSAE + CMI-VAE | 0.8000 | **87.47** | **88.0** |
| DSAE + $\beta$-VAE | 0.9000 | **87.20** | **87.6** |
| DSAE + CMI-VAE | 0.9000 | 87.00 | **87.6** |
| DSAE + $\beta$-VAE | 0.9900 | 87.33 | 87.6 |
| DSAE + CMI-VAE | 0.9900 | **87.60** | **88.6** |
| DSAE + $\beta$-VAE | 0.9990 | 87.27 | 87.4 |
| DSAE + CMI-VAE | 0.9990 | **87.60** | **88.0** |
| DSAE + $\beta$-VAE | 0.9999 | 86.87 | 87.2 |
| DSAE + CMI-VAE | 0.9999 | **87.73** | **88.0** |

labels (male or female) $\{y_i\}_{i=1}^{500}$, where each of the 2 classes had 250 samples. Then, we randomly created 250 pairs $\{(x_i, x_j)|y_i \neq y_j\}$, i.e., each pair consists of one male and one female speech. Using the trained DSAE, we encoded each $x$ into $z$ and $s$, created the pairs $\{(z_i, s_i, z_j, s_j)|y_i \neq y_j\}$, and decoded $z_i$ and $s_j$ ($z_j$ and $s_i$) to obtain $\hat{x}_{i,j}$ ($\hat{x}_{j,i}$), which ideally has the speaker characteristics of $x_i$ and the linguistic contents of $x_j$. Thus, we obtain 500 generated samples, where each $\hat{x}_{i,j}$ was labeled with $y_i$ assuming that the characteristics that tend to depend on gender (such as pitch) were successfully converted. Finally, we trained a logistic classifier with the 500 pairs $\{(\hat{x}_{i,j}, y_i)\}$ and evaluated the performance on real test speeches. Note that, because the raw $\hat{x}_{i,j}$ has an excessively high dimension ($20(T) \times 201$(features)) for the logistic classifier, $\hat{x}$ was averaged over the time-axis prior to its measurement. Intuitively, when the decoder ignores $z$, the generated samples might belong to a class different from the original ones, which produces label errors. Therefore, to achieve a high VC-mCAS, $z$ should capture global information but $s$ should not. Aos, the generated samples should be realistic to reduce the domain gap between train (generated) and test (real) data.

Table 2 presents the values of VC-mCAS for the objectives of $\beta$-VAE and CMI-VAE. Note that we report the mean and best scores within three random seed trials for each $\gamma$. The table illustrates that given a fixed $\gamma$, CMI-VAE nearly consistently achieved a higher VC-mCAS compared to $\beta$-VAE, indicating that regularizing $I'(z;s)$ is complementary to $\beta$-VAE. Furthermore, although $\gamma = 0.8$ yields a higher EER($z$) than those with $\gamma = 0.4$ in Table 1, it yields a higher VC-mCAS. Therefore, in addition to measuring EER, as was done in previous studies (Hsu et al., 2017; Yingzhen & Mandt, 2018), we claim that it is necessary to consider the performance of the controlled generation for evaluating the usefulness of the global representation.

## K   DETAILED EXPERIMENTAL RESULTS

### K.1   DETAILED EXPERIMENTAL RESULTS FOR DSAE

Table 3 presents the ELBO, KL, Recon, EER, and $\hat{I}(z;s)$ values of DSAE on TIMIT corpus. Regarding the estimation of $\hat{I}(z;s)$, please refer to Appendix A. Also, note that KL($z$) approximates $I(x;z)$ because it upper bounds $I(x;z)$, and has been used for the metric to assess whether a decoder ignores $z$ or not (Bowman et al., 2016; Alemi et al., 2018; He et al., 2019). Here, the ELBO, Recon, and KL($S$) values are not divided by $T = 20$. Also, the Recon values can be nevative because the variance of our decoder are learnable parameters.

As shown in the table, (i) given a fixed $\gamma$, the two methods ($\beta$-VAE and CMI-VAE) have the same level of KL($z$); therefore, both the methods can be used to alleviate PC. (ii) On the other hand, given a fixed $\gamma$, CMI-VAE achieved the lower $\hat{I}(z;s)$ values in most cases, suggesting that it facilitates the learning of good global representation. (iii) Finally, we have confirmed that even for large $\gamma$, there is still reasonable reconstruction performance for the both methods.

Table 3: The ELBO, KL, Recon, EER, and $\hat{I}(z; s)$ values of DSAE on TIMIT corpus. Each model was trained with a weighting parameter $\gamma$. "se" denotes standard error.

| Model | $\gamma$ | ELBO mean | se | KL(z) mean | se | KL(s) mean | se | Recon mean | se | EER(z) mean | se | EER(s) mean | se | $\hat{I}(z;s)$ mean | se |
|---|---|---|---|---|---|---|---|---|---|---|---|---|---|---|---|
| DSAE | 0.00 | 6299.08 | 14.06 | 18.00 | 0.09 | 495.92 | 4.45 | -6813.00 | 13.37 | 11.01 | 0.52 | 18.64 | 1.04 | 1.61 | 0.04 |
| DSAE + $\beta$-VAE | 0.40 | 6289.48 | 11.72 | 53.28 | 0.97 | 483.40 | 5.73 | -6826.16 | 15.46 | 3.88 | 0.15 | 29.45 | 0.29 | 2.60 | 0.01 |
| DSAE + CMI-VAE | 0.40 | 6288.64 | 4.82 | 54.13 | 2.51 | 468.06 | 14.03 | -6810.83 | 16.04 | 3.43 | 0.26 | 30.96 | 0.81 | 2.44 | 0.03 |
| DSAE + $\beta$-VAE | 0.80 | 6228.58 | 4.81 | 145.88 | 0.45 | 430.73 | 7.19 | -6805.20 | 10.76 | 4.33 | 0.26 | 38.84 | 0.47 | 2.84 | 0.05 |
| DSAE + CMI-VAE | 0.80 | 6222.22 | 4.60 | 145.09 | 0.17 | 430.59 | 5.60 | -6797.90 | 1.53 | 3.99 | 0.29 | 41.30 | 0.66 | 2.83 | 0.04 |
| DSAE + $\beta$-VAE | 0.90 | 6172.96 | 3.83 | 202.52 | 1.34 | 432.20 | 3.88 | -6807.67 | 2.13 | 4.55 | 0.15 | 39.42 | 1.06 | 2.89 | 0.01 |
| DSAE + CMI-VAE | 0.90 | 6192.05 | 6.58 | 199.89 | 1.10 | 429.53 | 7.84 | -6821.46 | 14.59 | 4.39 | 0.23 | 41.25 | 2.10 | 2.70 | 0.03 |
| DSAE + $\beta$-VAE | 0.99 | 6019.10 | 11.52 | 364.71 | 2.08 | 433.59 | 9.40 | -6817.40 | 19.39 | 6.33 | 0.34 | 38.63 | 1.18 | 3.27 | 0.15 |
| DSAE + CMI-VAE | 0.99 | 6031.71 | 5.78 | 361.03 | 1.85 | 434.55 | 8.67 | -6827.29 | 12.56 | 5.06 | 0.23 | 40.08 | 1.09 | 2.85 | 0.07 |

## K.2 DETAILED EXPERIMENTAL RESULTS FOR PIXELCNN-VAE

Tables 4 and 5 present the ELBO, KL($z$), Recon, $\hat{I}(z; s)$, mCAS, and AoLR values of PixelCNN-VAEs on MNIST and Fashion-MNIST. Moreover, these tables present mCAS(SVM) and AoSVM, which are the same with mCAS and AoLR except for using a support vector machine (SVM) with RBF kernel, i.e., a more powerful non-linear classifier, instead of the logistic classifier. Regarding the estimation of $\hat{I}(z; s)$, please refer to Appendix A. Also, note that KL($z$) approximates $I(x; z)$ because it upper bounds $I(x; z)$, and has been used for the metric to assess whether a decoder ignores $z$ or not (Bowman et al., 2016; Alemi et al., 2018; He et al., 2019). Here, the ELBO and Recon values are not divided by $T = 28 \times 28$.

As shown in the tables, (i) given a fixed $\gamma$, the three methods ($\beta$-VAE, MI-VAE, and CMI-VAE) have the same level of KL($z$); therefore, all the methods can be used to alleviate PC. (ii) On the other hand, given a fixed $\gamma$, CMI-VAE achieved the lower $\hat{I}(z; s)$ values in most cases, suggesting that it facilitates the learning of good global representation. (iii) Finally, even if we used a non-linear classifier SVM to calculate mCAS(SVM) and AoSVM, CMI-VAE achieved competitive or higher performance than the baselines in most cases. Note that, the exception is that given a $\gamma > 0.4$, there were not much differences in mCAS(SVM) for Fashion-MNIST within the three methods. One possible reason is that using the non-linear classifier increases the number of factors to be considered, such as overfitting, and makes fair comparisons difficult. Also, we note that using a very large $\gamma$ for PixelCNN-VAEs might not be a good idea. It is because when $\gamma$ becomes too large, the decoder of PixelCNN tends to resemble an identity mapping from $z$ to its output, regardless of the regularization method (e.g., see, generated samples for $\gamma = 0.6$ in Appendix K.3). To improve performance while avoiding this phenomenon, it could be useful to using a weighting parameter $\alpha > 1$ in Eq. 8 (e.g., using $\gamma = 0.3$ and $\alpha > 1$), and this remains an issue to be addressed in a future work as noted in Section 6.

## K.3 SAMPLE IMAGES FOR PIXELCNN-VAE

Figures 5 present the generated images with PixelCNN-VAEs.

Table 4: The ELBO, KL($z$), Recon, mCAS, AoLR, mCAS(SVM), and AoSVM values of PixelCNN-VAEs on MNIST. Each model was trained with a weighting parameter $\gamma$.

| $\gamma$ | Model | ELBO | KL($z$) | Recon | mCAS | AoLR | mCAS(SVM) | AoSVM | $\hat{I}(z;s)$ |
|---|---|---|---|---|---|---|---|---|---|
| 0.0 | $\beta$-VAE | 56.21 | 3.60 | 52.61 | 0.3966 | 0.6094 | 0.5722 | 0.6447 | 1.81 |
| 0.1 | $\beta$-VAE | 56.28 | 5.33 | 50.95 | 0.5917 | 0.8161 | 0.7172 | 0.8458 | 2.11 |
|  | CMI-VAE | 56.24 | 4.80 | 51.43 | 0.5421 | 0.7243 | 0.6499 | 0.7809 | 1.97 |
|  | MI-VAE | 56.23 | 5.16 | 51.07 | 0.5325 | 0.7602 | 0.6651 | 0.8080 | 2.05 |
| 0.2 | $\beta$-VAE | 56.68 | 9.21 | 47.47 | 0.6924 | 0.8479 | 0.7750 | 0.8980 | 2.45 |
|  | CMI-VAE | 56.68 | 9.32 | 47.36 | 0.7229 | 0.8692 | 0.7934 | 0.9026 | 2.29 |
|  | MI-VAE | 56.50 | 8.90 | 47.60 | 0.6734 | 0.8208 | 0.7606 | 0.8762 | 2.38 |
| 0.3 | $\beta$-VAE | 58.33 | 18.04 | 40.29 | 0.7448 | 0.8354 | 0.8109 | 0.9027 | 2.85 |
|  | CMI-VAE | 58.03 | 17.52 | 40.51 | 0.7716 | 0.8630 | 0.8136 | 0.9182 | 1.80 |
|  | MI-VAE | 57.81 | 16.37 | 41.44 | 0.7399 | 0.8238 | 0.8014 | 0.8938 | 2.85 |
| 0.4 | $\beta$-VAE | 61.22 | 29.00 | 32.21 | 0.7476 | 0.8204 | 0.8063 | 0.8962 | 3.28 |
|  | CMI-VAE | 61.27 | 29.56 | 31.71 | 0.7664 | 0.8401 | 0.8122 | 0.9114 | 2.21 |
|  | MI-VAE | 60.52 | 27.10 | 33.43 | 0.7461 | 0.7903 | 0.7985 | 0.8742 | 3.11 |
| 0.5 | $\beta$-VAE | 64.55 | 37.79 | 26.76 | 0.7555 | 0.8279 | 0.8060 | 0.9010 | 3.16 |
|  | CMI-VAE | 64.69 | 38.40 | 26.28 | 0.7575 | 0.8405 | 0.8075 | 0.9088 | 1.79 |
|  | MI-VAE | 63.57 | 35.62 | 27.95 | 0.7454 | 0.7719 | 0.7963 | 0.8582 | 3.18 |
| 0.6 | $\beta$-VAE | 68.74 | 46.29 | 22.45 | 0.7642 | 0.8306 | 0.8120 | 0.9056 | 3.27 |
|  | CMI-VAE | 69.00 | 47.11 | 21.90 | 0.7618 | 0.8486 | 0.8079 | 0.9190 | 2.57 |
|  | MI-VAE | 67.72 | 44.16 | 23.57 | 0.7484 | 0.7722 | 0.7949 | 0.8572 | 3.19 |
| 0.7 | $\beta$-VAE | 74.47 | 56.04 | 18.43 | 0.7531 | 0.8454 | 0.7984 | 0.9148 | 3.27 |
|  | CMI-VAE | 74.71 | 56.28 | 18.43 | 0.7553 | 0.8486 | 0.8019 | 0.9179 | 1.81 |
|  | MI-VAE | 72.92 | 52.88 | 20.05 | 0.7496 | 0.7753 | 0.7912 | 0.8601 | 3.19 |

Table 5: The ELBO, KL($z$), Recon, mCAS, AoLR, mCAS(SVM), and AoSVM values of PixelCNN-VAEs on Fashion-MNIST Each model was trained with a weighting parameter $\gamma$.

| $\gamma$ | Model | ELBO | KL($z$) | Recon | mCAS | AoLR | mCAS(SVM) | AoSVM | $\hat{I}(z;s)$ |
|---|---|---|---|---|---|---|---|---|---|
| 0.0 | $\beta$-VAE | 88.60 | 4.02 | 84.58 | 0.4820 | 0.6906 | 0.6342 | 0.7552 | 2.00 |
| 0.1 | $\beta$-VAE | 88.86 | 6.71 | 82.16 | 0.5716 | 0.7172 | 0.6556 | 0.7790 | 2.40 |
|  | CMI-VAE | 88.90 | 6.47 | 82.43 | 0.5837 | 0.7201 | 0.6618 | 0.7757 | 2.53 |
|  | MI-VAE | 88.84 | 6.72 | 82.12 | 0.5645 | 0.7166 | 0.6592 | 0.7742 | 2.37 |
| 0.2 | $\beta$-VAE | 89.94 | 10.95 | 78.99 | 0.6174 | 0.7092 | 0.6792 | 0.7708 | 2.78 |
|  | CMI-VAE | 90.25 | 11.49 | 78.76 | 0.6463 | 0.7258 | 0.6885 | 0.7779 | 2.46 |
|  | MI-VAE | 89.78 | 10.55 | 79.23 | 0.6132 | 0.6960 | 0.6760 | 0.7655 | 2.69 |
| 0.3 | $\beta$-VAE | 91.45 | 16.11 | 75.34 | 0.6427 | 0.7030 | 0.6811 | 0.7665 | 2.94 |
|  | CMI-VAE | 91.55 | 16.39 | 75.16 | 0.6578 | 0.7238 | 0.6873 | 0.7713 | 2.03 |
|  | MI-VAE | 91.14 | 15.09 | 76.06 | 0.6241 | 0.6817 | 0.6897 | 0.7507 | 2.92 |
| 0.4 | $\beta$-VAE | 93.98 | 24.33 | 69.65 | 0.6536 | 0.7062 | 0.6858 | 0.7636 | 3.24 |
|  | CMI-VAE | 93.81 | 23.35 | 70.46 | 0.6662 | 0.7145 | 0.6868 | 0.7701 | 2.11 |
|  | MI-VAE | 93.36 | 22.38 | 70.98 | 0.6366 | 0.6787 | 0.6886 | 0.7278 | 3.21 |
| 0.5 | $\beta$-VAE | 97.36 | 33.02 | 64.34 | 0.6547 | 0.6919 | 0.6845 | 0.7521 | 3.36 |
|  | CMI-VAE | 97.14 | 32.14 | 65.00 | 0.6651 | 0.7193 | 0.6867 | 0.7688 | 2.29 |
|  | MI-VAE | 96.47 | 31.19 | 65.27 | 0.6343 | 0.6506 | 0.6823 | 0.7056 | 3.31 |
| 0.6 | $\beta$-VAE | 101.76 | 42.40 | 59.36 | 0.6633 | 0.6899 | 0.6853 | 0.7524 | 3.39 |
|  | CMI-VAE | 101.83 | 41.97 | 59.86 | 0.6685 | 0.7112 | 0.6904 | 0.7659 | 2.43 |
|  | MI-VAE | 100.66 | 39.96 | 60.70 | 0.6535 | 0.6420 | 0.6854 | 0.6961 | 3.33 |
| 0.7 | $\beta$-VAE | 107.77 | 52.57 | 55.20 | 0.6680 | 0.6983 | 0.6900 | 0.7545 | 3.41 |
|  | CMI-VAE | 107.49 | 51.97 | 55.52 | 0.6729 | 0.7103 | 0.6910 | 0.7681 | 2.00 |
|  | MI-VAE | 106.05 | 49.14 | 56.91 | 0.6578 | 0.6278 | 0.6838 | 0.6791 | 3.34 |
| 0.8 | $\beta$-VAE | 117.00 | 65.76 | 51.25 | 0.6649 | 0.7044 | 0.6855 | 0.7665 | 3.37 |
|  | CMI-VAE | 116.21 | 64.68 | 51.53 | 0.6634 | 0.7098 | 0.6851 | 0.7705 | 2.35 |
|  | MI-VAE | 113.98 | 60.19 | 53.79 | 0.6599 | 0.6108 | 0.6890 | 0.6709 | 3.31 |
| 0.9 | $\beta$-VAE | 132.62 | 85.13 | 47.49 | 0.6695 | 0.7097 | 0.6844 | 0.7787 | 3.47 |
|  | CMI-VAE | 131.01 | 82.97 | 48.04 | 0.6703 | 0.7169 | 0.6860 | 0.7780 | 2.74 |
|  | MI-VAE | 128.34 | 77.72 | 50.61 | 0.6565 | 0.6209 | 0.6880 | 0.6882 | 3.31 |

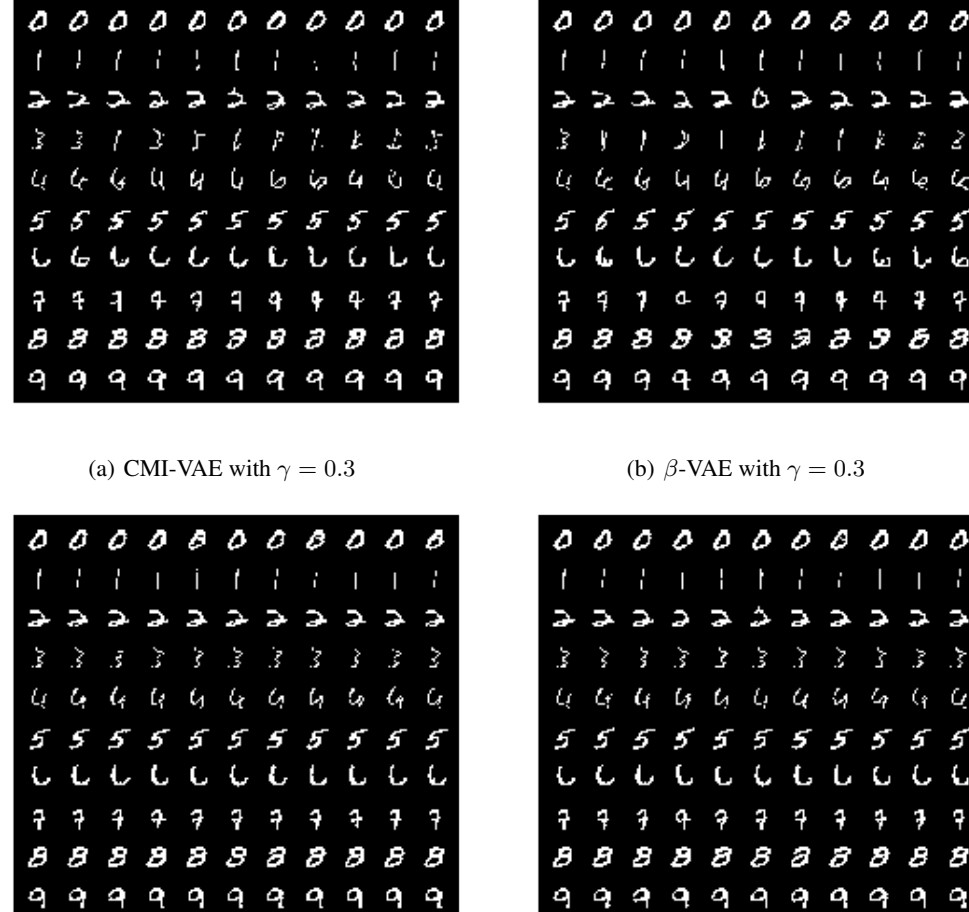

(a) CMI-VAE with $\gamma = 0.3$       (b) $\beta$-VAE with $\gamma = 0.3$

(c) CMI-VAE with $\gamma = 0.6$       (d) $\beta$-VAE with $\gamma = 0.6$

Figure 5: Real images (the first column) and generated images by PixelCNN-VAEs (the other 10 columns). The images in each row are stochastically sampled from the decoder $p(x|z)$ using the same $z$, which is extracted from $x$ in the first column. The figures present that the diversity of the images in $\gamma = 0.3$ is better than that in $\gamma = 0.6$, which may be because PixelCNN-VAE would resemble an identity mapping with a large $\gamma$. In contrast, $\gamma = 0.3$ apparently produces more label errors than $\gamma = 0.6$ because the decoder ignores $z$ with a small $\gamma$ (see, e.g., the rows for 3 and 4). Furthermore, when comparing (a) (CMI-VAE with $\gamma = 0.3$) and (b) ($\beta$-VAE with $\gamma = 0.3$), apparently, (a) produces less label errors (see, e.g., the rows for 2 and 3). This result is consistent with the mCAS scores in Figure 3 (Section 5.3), which indicates that CMI-VAE achieved better diversity and less label errors than $\beta$-VAE.

