# OpenReview forum: "Information Theoretic Regularization for Learning Global Features by Sequential VAE"
_ICLR.cc/2021/Conference — Reject_

### Official Review · AnonReviewer3 · 2020-10-27

**Rating:** 6
**Confidence:** 4

**Review:**

Update after rebuttal:
I agree with Reviewer 5 that this paper has good ingredients, and the discussion and update of the draft clarifies the novelty and provides better review on the related work. However, the experiments presented in this paper are not very comprehensive, particularly the baselines and the ablation/alternative studies. I am not fully convinced by the authors response of "because MI-VAE performed worse than $\beta$-VAE in PixelCNN-VAEs ... we expected that a similar tendency would be observed." PixelCNN-VAE uses an autoregressive decoder, which are known to exhibit issues that are not observed from non-autoregressive ones like DSAE. This explanation of why MI-VAE was emitted seems slightly hand-wavy. I decided to decrease the rating to 6 to reflect the insufficiency in experiments, but hope that this experiment can be added if the paper is accepted.

Summary:
This paper aims to learn representations that capture global features in structured data, such as the speaker information within speech or the digit class in an image. The authors argue that previous work regularizing the representation $z$ by maximizing its mutual information $I(x; z)$ with the data $x$ has the side effect of simultaneously maximizing the mutual information between $z$ and local features $s$. This may cause the global feature $z$ to encode unwanted local information or vice versa. To address this issue, the authors propose to regularize $z$ through maximizing $I(x;z) - I(z;s)$, which is a lower bound of the conditional MI $I(x;z|s)$. The proposed regularization is further estimated using the density ratio trick, which employs a discriminator to estimate the ratio $\dfrac{q(z,s)}{p(z)q(s)}$ via a binary classification task and provide training signals to the encoder. The proposed regularization is applied to DSAE and PixcelCNN-VAE to demonstrate its effectiveness.

Pros:
- The paper is well written and easy to follow. Motivations of this work are clear, and related studies are also properly described to contrast the difference from this work.
- The formulation of the regularization seems novel to me. Derivation and approximations also make sense.
- Experiments conducted on multiple domains (speech and images) demonstrate superior performance compared to the baseline methods (no regularization or MI-based regularization). Multiple metrics are adopted for evaluation of different aspects (e.g., AoLR/EER evaluates linear separability, mCAS indirectly evaluates diversity)

Cons/Questions:
- The experiments on images compared the proposed method with both beta-VAE and MI-VAE. I am curious why the authors only compare with beta-VAE for the speech experiments but not MI-VAE. The EER reported in Hsu et al. (2017) is much lower than the results in this paper, and as mentioned by the authors, that work regularizes the representations with a discriminative objective that approximates $H(x|z)$ and therefore can be seen as a form of MI-VAE. The authors should also compare with such regularization for the speech experiments since it’s shown effective in the previous work.
- The authors state that approximating both $I(x;z)$ and $I(z;s)$ may complicate optimization, and avoid doing so by forcing alpha in Eq.8 to be 1 to enable rearrangement in Eq.10. However, it would be informative to show the results of approximating both terms with the density ratio trick and see how much it would affect the performance. By doing so, the strength of regularizing $I(z;s)$ can be independently tuned.

---

> ### Author Response · Authors · 2020-11-20
> **Author Response to AnonReviewer3**
>
> We thank the reviewer for taking the time to thoroughly read our paper and giving us the opportunity to strengthen our manuscript with your valuable comments and queries. Below we respond to concerns that the reviewer commented.
>
> **A1 Regarding "why the authors only compare with $\beta$-VAE for the speech experiments but not MI-VAE."**
> Thank you for your query. Because MI-VAE performed worse than $\beta$-VAE in our experiments of PixelCNN-VAEs, which may be due to the adversarial training in MI-VAE causing optimization difficulties (see, Section 5.3), we expected that a similar tendency would be observed in the experiment using speech data.
>
> **A2 Regarding "The authors should also compare with such regularization for the speech experiments since it’s shown effective in the previous work"**
> Thank you for your suggestion.
>
> (point 1) Firstly, "MI-VAE" used in our experiment is different from the method of Hsu et. al. (2017), i.e., discriminative objective. Also, we did not compare with MI-VAE due to the reason explained in A1.
>
> (point 2) Secondly, we agree that we should have noted that "The EER reported in Hsu et al. (2017) is much lower than the results in this paper"; however, we believe that even if our method cannot outperform discriminative objective, our claim, "regularizing $I(x; z)$ and $I(z; s)$ is complementary", can be defended. It is because we compared CMI-VAE (regularization of $I(x; z)- I(z; s)$) to $\beta$-VAE (regularization of $I(x; z)$), and showed that CMI-VAE consistently outperformed $\beta$-VAE.
>
> (point 3) Thirdly, we chose to extend $\beta$-VAE to construct the proposed objective function (Eqs. 13 and 14), although the discriminative objective could also be extended to CMI-regularization by the addition of the $I(z; s)$ minimization term as noted in Section 6. It is because we believe that $\beta$-VAE is the simplest MI-maximization method that requires fewer hyperparameters, widely used in (sequential) VAE community (e.g., [1]).
>
> Then, we have added the discussion about point 2 to the last paragraph in Section 5.2, and added the discussion about point 3 to Section 6.
>
> **A3 Regarding "it would be informative to show the results of approximating both terms ... can be independently tuned."**
> Thank you for your suggestion. We agree that the experiments would be informative; however, since we cannot afford to conduct it within the rebuttal period, we have added it as a future work to Section 6. Also, while independent tuning would be informative, we believe that "forcing alpha in Eq.8 to be 1" has two merits as noted in Section 3.2 and Appendix D: it not only avoids complicating optimization, but also reveals the relation between our regularization term and conditional mutual information.
>
> Again, thank you for giving us the opportunity to strengthen our manuscript with your valuable comments and queries.
> We look forward to hearing from you regarding our submission.
>
> [1] He et. al., Lagging inference networks and posterior collapse in variational autoencoders. ICLR, 2019.

---

### Official Review · AnonReviewer2 · 2020-11-01
**Good paper, accept**

**Rating:** 7
**Confidence:** 4

**Review:**

Summary
Observing the deficiency of existing MI-based sequential VAEs, where through the learning objective design, the local and global features may become disentangled, the authors propose adding a regularizer to explicitly disentangle local and global features. To address the computation tractability issue of adding the new regularizer, a density-ratio based approximation is adopted and a classifier is trained to approximate the density-ratio term, which during training is learned alternatively with the VAE objective.
Relation between the proposed method and Conditional MI, beta-VAE, and domain adversarial training is discussed. Then the proposed method is empirically evaluated on two sequential VAE based tasks: speech voice manipulation (DSAE) and image generation (PixelCNN-VAE). The authors designed experiments that show that the proposed approach can improve global representation learning.

Quality

*Pros:
The paper is overall of good quality: the context of the problem is well explained with adequate diagrams/plots to aide understanding. The approach is well motivated and the derivation of the approximation method is mostly easy to follow. Experiments are well designed and details are provided.

*Cons:
The density-ratio technique to approximate D(q(z, s) || p(z)q(s)) should be better explained in the paper: I had to look into the reference (Sugiyama et al 2012) to understand the derivation in Equation (11).

Clarity
The paper is clearly written.

Originality
The paper is based on existing work of MI-VAE families of neural networks. Instead of proposing a completely new architecture/method, the authors spot a gap in the current literature, i.e., that local and global feature representations can be disentangled using the current learning objective. The paper addresses exactly this gap.

Significance
The paper and the proposed method should be of some significance to the VAE and domain-adaptation community and inspire future works.

---

> ### Author Response · Authors · 2020-11-20
> **Author Response to AnonReviewer2**
>
> We thank the reviewer for taking the time to thoroughly read our paper and giving us the opportunity to strengthen our manuscript with your valuable comments and queries. Below we respond to a concern that the reviewer commented.
>
> **Regarding "The density-ratio technique ... should be better explained in the paper"**
> Thank you for your suggestion. We have reflected this comment by two ways. Namely, (i) we have added a brief explanation of the density-ratio trick (DRT) (see, Sec.3.2). Also, (ii) we have added a citation [1], which discusses the relationship between DRT and adversarial training and can further facilitate the understanding (see, Sec.3.2).
>
> Again, thank you for giving us the opportunity to strengthen our manuscript with your valuable comments and queries. We look forward to hearing from you regarding our submission.
>
> [1] Shakir Mohamed and Balaji Lakshminarayanan. Learning in implicit generative models. 2017.

---

### Official Review · AnonReviewer5 · 2020-11-07
**Good main idea, but hard to judge whether approximations are justified and whether method would work well on more complex data**

**Rating:** 6
**Confidence:** 3

**Review:**

**Update after discussion with authors**
I want to thank the authors for their incredibly detailed responses and engaging so actively in the discussion. Some of my criticism could be addressed, while other issues are still somewhat open. If the main merit of the paper is to make the "sequential VAE community" aware of issues that have been discussed and addressed before, then I think the paper does an OK job at that (though I'm not entirely sure what community that is, the issues have been discussed before in the fields of vision and VAEs with autoregressive decoders).

I want to strongly encourage the authors to be as precise as possible when describing the novelty - maximizing the mutual information that a representation carries w.r.t. some relevance variable while simultaneously minimizing information that it carries w.r.t. to another variable is NOT novel. What is novel is the application of that principle to separating "global" from "local" information in sequential data (and how to actually perform this originally intractable optimization in practice). I also want to encourage the authors to state what's known and what's new as clearly as possible and improve the quality and clarity of the "educational" review of why maximizing mutual information is not enough as much as possible.

Viewing the paper as "showing how a known problem also appears when separating global from local information, and how to apply known solution-approaches to the problem in this specific context", shifts the relative importance of the issues raised by me. Essentially, that view emphasizes the paper as mostly an application paper (rather than a novel theoretical contribution). Accordingly (but please make sure that that shift in view is also clear in the final paper), I am weakly in favor of accepting the paper and have updated my score accordingly.

---

**Summary**
The paper tackles the problem of separating ‘global’ from ‘local’ features in unsupervised representation learning. In particular, the paper tackles a common problem in autoencoders where the decoder (generative model) is autoregressive and conditioned on a variable. Ideally, the latter variable captures all global information (such as e.g. speaker identity) whereas the autoregressive model deals with generating local structure (such as e.g. phonemes). As the paper points out, capturing only global information (and all of it) in the conditioning variable alone is notoriously difficult with standard variational autoencoder objectives, and several solutions have been proposed in the past. In this paper the idea is to add an explicit penalty for statistical dependence (mutual information) between the global and the local random variable. This intractable objective is simplified with a series of approximations, leading to a novel training objective. Results are shown for speech data, and MNIST/FashionMNIST, where the proposed training objective outperforms a beta-VAE objective and an objective that explicitly aims to maximize information on the global variable.

---
**Contributions, Novelty, Impact**

1) Analysis of shortcomings of mutual-information maximization to regularize latent representations into capturing ‘global’ features. This topic has been widely discussed in the literature before, typically in the context separating nuisance factors from relevant variations in the data, or more broadly: separating relevant from irrelevant information (which is canonically addressed in the information bottleneck framework of course). Most of this previous discussion was aimed at supervised learning ([2]), but there is a considerable body of work in unsupervised learning as well ([1] discusses the same issue but with more clarity), and some recent, very relevant work targeting VAEs with autoregressive decoders as well ([3] is among the state-of-the-art models). The paper provides a recap of this literature, but misses some key references, and the clarity of the writing (pages 1-4) could be improved (see my comments on clarity below). Therefore I would rate the impact of this contribution as low.

2) Proposal of a novel regularizer. The main idea behind the regularizer Eq. (8) is good, but certainly not novel - it has been broadly discussed in the literature and implemented in various ways. The merit is thus in the particular derivation and approximations that lead to the objective in Eq. (13) and (14). To me the derivation seems correct, though the precise motivation is somewhat unclear (what shortcomings of alternative approaches are addressed here, e.g. using the density ration trick?). I personally think that there is sufficient novelty, but in the current manuscript it is hard to assess whether the novel method has benefits compared to strong competitor methods (which are unfortunately missing from the experiments).

3) Experiments on a speech dataset (using a state-space-model decoder), and MNIST/FashionMNIST (using a PixelCNN). Results indicate that the extracted latent space does capture global features slightly better than a beta-VAE, or (quite a bit better than) a MI-maximizing VAE. There is also some indication that local features capture less global information with the proposed method compared to a beta-VAE. These results are promising, but not surprising since beta-VAE and MI-VAE were not designed to solve the shortcomings that the method is trying to address. For results to be more convincing and stronger, it would be good to compare against alternative approaches that have the same objective, such as e.g. [1] and [3]. Additionally more control experiments and ablations as well as reporting more metrics (l(x;z) and I(z;s), or proxies/approximations thereof) would strengthen the findings and thus the potential impact (see my comments on improvements below).

(I am not an author of any of these)
[1] Moyer et al. Invariant Representations without Adversarial Training, 2018
[2] Jaiswal et al. Discovery and Separation of Features for Invariant Representation Learning, 2019
[3] Razavi et al. Generating Diverse High-Fidelity Images with VQ-VAE-2, 2019

---
**Score and reasons for score**
The paper addresses an important problem that has received attention in the literature for at least two decades (the InfoBottleneck framework lays the theoretical foundations here). The particular application to: (i) unsupervised learning, and (ii) global-conditioned autoregressive (VAE) models is very timely and has received less attention in the literature (but there are some papers).

My main issue is that the paper addresses two problems: (A) separating global from local information, (B) avoiding that autoregressive decoders ignore the global latent variable. Clearly stating both problems, reviewing the literature for each of them, and then showing how the paper solves both of them (and showing experimental results for both of them) would really help with clarity and readability of the paper. It would also help flesh out the novel contributions made by the paper. Additionally, it is not entirely clear how well the proposed objective actually addresses (A) and (B) in the experiments. There is some good indication for (A), but it is not directly measured (e.g. by estimating I(x:z) and I(s;z)), the effect is only shown indirectly via AoLR and mCAS (or Err(z) and Err(s)). The same is true for (B): there is some that the generative model does not ignore the global latent code via the mCAS experiments, but it is quite indirect (also looking at the generated examples in appendix K raises some doubts about diversity of the generative model).

Overall I think the ingredients for a good paper are there, but they are not quite coming together yet. A deeper look into the empirical results (control experiments, additional metrics), and a comparison against strong competitor methods are needed. My recommendation would also be to really focus on the new objectives (Eq. 13 and 14) and discuss in more detail how they differ from competitor approaches and what the theoretical/empirical advantages of these differences are (for instance I am personally not yet fully convinced that using the density-ratio-trick with a neural network classifier will always work well in practice). If all of that were in place, I think the paper would be significantly stronger and could potentially have wide impact. I thus recommend a major revision of the work, which is not possible within the rebuttal period. Below are concrete suggestions for improvements, and I will of course take into account the other reviews and authors’ response.

---
**Strengths**

1) The problem (separation of global and local info in variational unsupervised representation learning with autoregressive decoders) is timely and important, and has been somewhat neglected in the representation learning community (though there is work out there, and the same problem has been discussed extensively in a related context, such as e.g. supervised learning).

2) The Method builds on a body of previous great work and applies it in an interesting context (global vs. local features).

---
**Weaknesses**

1) Merits of the method somewhat unclear - the motivation/derivation when going from Eq. 8 to Eq. 13, 14  is a bit ad-hoc. What alternatives are there to the choices/approximations made? What are the advantages/disadvantages of these? Answering this might also involve some control experiments and ablations.

2) The experiments show somewhat indirectly that the goals were achieved. There is some empirical evidence that the method is working to some degree, but it remains unclear whether e.g. the learned z capture only global information (and all of it), and whether s captures only local information (and how much of it). What’s needed here are additional results/experiments.

3) The current writing is ok but could be improved. I think it would be helpful to clearly state the problems and previous approaches of solving them (and the issues with these previous approaches). This would make it easier to see how the proposed method fits into the wider picture and which specific problem it addresses/improves upon. I think Alemi 2018 (cited in the paper) and [1] mentioned above do a very good job of describing the overall problem..

---
**Correctness**
The derivation of the method looks ok to me, though it would be nice to justify the approximations made and attempt to empirically verify that they do not have a severe impact on the solution. The conclusions drawn from the experiments are broadly ok, but since the evaluation measures the desired properties in a quite indirect way, the generality of the findings and the extent to which the method solves the problem (quantitatively) remain somewhat unclear.

---
**Clarity**
It took me a bit longer to follow the main line of arguments than it should have (which might of course be my fault). It’s a bit hard to pinpoint to a specific paragraph, but perhaps the following suggestions are helpful for improving readability. It might be worth clearly stating the main problems (denoted (A) and (B) further up in ‘Score and Reasons for Score’) and separately discussion how they have been addressed (and what the remaining issues are) and how the paper addresses them. Currently this is entangled in the derivation of the method.

It would probably also help to have a short paragraph that summarises the novel contributions clearly (which makes it clear what’s novel and what’s been proposed before).

---
**Improvements (that would make me raise my score) / major issues**

1) Comment on all assumptions made when going from Eq (8) to (13), (14). Are these assumptions justified in practice? Would there be alternative choices, and if yes what are the downsides of these alternative choices? Some of the assumptions will then lead to further control experiments that would ideally be included in the paper. One example (perhaps the most important one) is below in 2)

2) Neural network classifiers are notoriously known to be ill-calibrated (typically having over-confident probability estimates). This could be problematic in the DRT approximation since the discriminator’s output probability crucially matters! Is the discriminator well calibrated in practice? How robust is the method against calibration issues? Is the problem expected to get worse when scaling to more complex data and bigger network architectures? This needs to be discussed, but ideally some points are also verified empirically.

3) beta-VAE and MI-VAE are ok baselines, but are not sufficient to show that the method performs very well. These two baseline methods have not been designed to address the main issue (separating global from local information) - it is thus not too surprising that the proposed method performs well. What’s needed is comparisons against strong baselines, e.g. a (hierarchical) VQ-VAE2 (ref [3] further up). Given that the method only slightly outperforms beta-VAE on the metrics shown (which has no explicit incentive to capture global information) this is important.

4) Report additional metrics (for each experiment it would be good to also report: reconstruction error, estimates of I(x;z) and I(z;s)). As \gamma is varied, does the method lead to consistent increase in I(x;z) and decrease in I(z;s)? Are the values for the latter two significantly better than when using beta-VAE/MI-VAE?

5) Reporting AoLR and mCAS with a logistic regressor/classifier is ok, because it says something about latent-space geometry which could be interesting. But for the paper it is more important to capture the exact amount of global information captured by z and s. Therefore it would be good to show additional results for AoLR and mCAS where the regressor/classifier is a powerful nonlinear model (a deep neural net).

6a) Alemi et al. 2018 gets cited quite a bit in the paper, but is not very well represented in the paper. In particular: the paper proposes a quantitative measure as a diagnostic to see how much information is captured by the latent variable and how much of that is used/ignored by the decoder (which leads to the definition of several operating regimes, such as the “auto-decoding” regime). Why not report the same measure in the current method?

6b) Alemi et al. 2018 actually propose a modified objective to target a specific rate. They empirically observe that a beta-VAE with beta<1 *in their experiments* leads to the VAE operating in the desired regime. As far as I understand they do not propose this as a general solution to fix decoders ignoring latent codes. This should be mentioned in the paper. As a consequence 6a) becomes even more important, or without any verification beta-VAE becomes an even weaker baseline, meaning that comparison against strong methods becomes more important.

---
**Minor comments**

a) Eq. (10) should be an inequality, because I(z;s) is upper bounded on r.h.s.?

b) How was it determined that “alpha=1 works reasonably”, is this based on some control experiments?

c)  Eq (13). Why this particular mixing in of the KL-term, why not multiply KL(s) with (1-\gamma) as well?

d) Table 1: report the reconstruction error. In particular, for high \gamma is there still reasonable reconstruction performance (and thus separation into global z and local s), or is all information except global information discarded and s essentially does not capture much meaningful information anymore, making good reconstruction impossible?

e) Fig 3a - is the x-axis ELBO or KL?

f) Fig 3, Table 1: ideally report multiple repetitions with error bars.

g) For \gamma=0.6 in appendix K, there seems to be very little diversity in samples drawn from either model. This should be mentioned more clearly in the main text.

---

> ### Author Response · Authors · 2020-11-20
> **Author Response to AnonReviewer5 (1/4)**
>
> We thank the reviewer for taking the time to thoroughly read our paper.
> We are glad to hear that the ingredients for a good paper are there, and have worked hard to incorporate your valuable feedback.
> We have separated our responses into three parts.
> In Part A, we respond to major concerns of the reviewer (denoted as A1 to A4).
> In Part B, we respond to each comments in the "Improvements / major issues" paragraph (B1 to B7).
> In Part C, we respond to each comments in the "Minor comments" paragraph (C1 to C7).
>
> ## Part A
>
> ### A1 Regarding our contribution
> Thank you for your suggestion that "It would probably also help ... summarises the novel contributions clearly".
> We agree with you, and we first clarify our contributions here.
>
> (i) Through our information-theoretic analysis, we reveal the potential negative side-effect of MI-maximizing regularization, which has been standard in learning global representation with sequential VAEs as discussed in Section 1, 2.2, and 4.
> This analysis makes the sequential VAE community aware of the limitation of the regularization, and encourages the community to seek for new regularization approach.
> Namely, our analysis formalises the following mechanism (a-c):
> - (a) Sequential VAEs with a global latent variable $z$ can in principle uncover global representation of data by exploiting its structured data generating process.
> - (b) Previous studies for the sequential VAEs have regularized mutual information $I(x; z)$ to be large in order to alleviate posterior collapse (PC).
> - (c) However, (b) can increase $I(z; s)$ as a side-effect, which contradicts the intention of (a).
>
> (ii) by analyzing the mechanism, we proposed a natural regularization approach that maximizes $I(x; z)$ and minimizes $I(z; s)$ *at the same time* in order to obtain *good global representation*.
> $I(x; z)$ and $I(x; z)$ are robustly shown to work complementary by our experiments using two models (DSAE and PixelCNN-VAE) and two domains (speech and image datasets).
> This finding would help improve the various sequential VAEs proposed before, which are presented in Section 4.
>
> To clarify our contribution, we have added the above discussion to the last paragraph of Section 1 (denoted as Update 1).
> Also, we have revised Section 3.1 to further clarify the mechanism (a-c) (Update 2).
>
>
> ### A2 Regarding "the main problems (denoted as (A) and (B))"
> Than you for your suggestion that we should clarify the following points.
>
> (i) Firstly, the reviewer suggests that "It might be worth clearly stating the main problems (denoted as (A) and (B))", and queries that "how the paper addresses them".
> You have raised an important point.
> We believe this point has already been clarified in A1.
> That is, our main problem is not to tackle (A) and (B) independently, but rather learn good representation via regularizing $I(x; z)$ and $I(z; s)$.
> The "good representation" is one that facilitates downstream applications such as controlled generation (e.g., voice conversion) and semi-supervised learning.
> Also, we do not consider (A) and (B) as independent issues;
> rather, our analysis reveals the relation betweein (A) and (B): (A) becomes problematic as a side-effect of (B) in the sequential VAEs.
>
> (ii) Secondly, the reviewer suggests that "separately discussion how they have been addressed".
> We have incorporated this suggestion.
> Namely, we have redrafted the 3rd paragraph of Section 4 to discuss "how (A) have been addressed" (Update 3).
> On the other hand, "how (B) have been addressed" is already discussed in the 2nd paragraph of Section 4 of the original manuscript.
>
> (iii) Thirdly, the reviwer queries that "it is not entirely clear how well the proposed objective actually addresses (A) and (B) in the experiments."
> We acknowledge that assessing (A) and (B) would improve our paper;
> therefore, we have reported the metrics suggested in "Improvements 4" to Appendix K1 and K2 (Update 4).
> However, we believe our contribution is defended with only the existing experiments because our purpose is not to increase $I(x; z)$ and decrease $I(z; s)$, but to obtain *good global representation* via regularizing them (see, A1);
> Therefore, *measuring representation quality* is a direct way.
> Moreover, EER and AoLR are the conventional metrics to assess the quality of global representation, e.g., Li et. al. and Razavi et. al. also use these metrics.
> On the other hand, large $I(x; z)$ value and small $I(z; s)$ value do not necessarily indicate that $z$ is a good global representation.
> (For example, imagine the situation where $z$ has all the information about $x$ and $s$ has no information about $x$.
> In this case, although $I(x; z)$ is high and $I(z; s)$ is low, $z$ does not capture only global information.)
> We have further clarified these points by redrafting the 1st paragraph of Section 5.1 (Update 5).

---

> > ### Comment · AnonReviewer5 · 2020-11-23
> > **Answer 1/4**
> >
> > I want to thank the authors for their thorough and detailed answer. I will answer to the points addressed separately. I  will update my review based on the responses and other reviews after thoroughly engaging with the material.
> >
> > A1: To anyone familiar with the relevant literature (a-c) should be no surprise. Perhaps it is not well known in the "sequential VAE community" (though it has been reported in computer vision when using autoregressive decoders in VAEs, a particular form of a sequential VAE). It's fine to point out an important known issue to make the wider community aware of the issue. But it is also important to clearly state the novel contributions and separate them from what's known and previously reported.
> >
> > The same is true for (ii) - the objective (Eq. 8 in the paper) has been proposed and used before in very similar contexts (typically not for separating out a global conditioning variable but for separating out other kinds of information, such as invariant features). The novelty here is thus the application to a specific problem (but not the analysis by decomposing multivariate mutual information). The more precise and clear the paper is about what's been known and done before and what's novel the better.
> >
> > A2: Thanks for going into more detail about the issues (A) and (B) - though they are coupled problems, each of the issues can be stated, and they each have received treatment in the previous literature individually. Re: (iii) I understand that the goal is not to simply maximize I(x;z) and minimize I(z;s). But the two quantities (together with other measurements, like reconstruction error and ELBO value) are important diagnostic quantities that are worth reporting (particulary when following the reasoning in "Fixing a broken ELBO"). Thanks for including them in the appendix.

---

> > > ### Author Response · Authors · 2020-11-24
> > > **Author Response (1/3)**
> > >
> > > We thank the reviewer for taking the time to thoroughly read our response and giving us the further feedback to strengthen our manuscript.
> > > We will respond to the points separately.
> > >
> > > **A1 "it is also important to clearly state the novel contributions and separate them from what's known and previously reported"**
> > > We agree with you, and have updated the 3rd and 4th paragraph of Section 4 to further clarify the difference between our paper and the previous studies.

---

> ### Author Response · Authors · 2020-11-20
> **Author Response to AnonReviewer5 (2/4)**
>
> ### A3 Regarding the novelty of Eq. (8)
>
> (i) The reviewer comments that "Eq. (8) is good, but certainly not novel .. The merit is thus in ... in Eq. (13) and (14)".
> You have raised an important point;
> however, our novelty is not "particular derivation and approximations" of Eq. (13) and (14), but is "the regularizer Eq. (8)".
> That is, as noted in A1, our novelty is to regularize $I(x; z)$ and $I(z; s)$ *at the same time*.
> It is novel, especially in the context of the sequential VAEs, due to two reasons.
> Firstly, in previous studies of the sequential VAEs, although the regularization of $I(x; z)$ was proposed, the regularization of $I(z; s)$ has been overlooked (see, also, the mechanism (a-c) in A1).
> Secondly, in the studies for separating relevant from irrelevant information (such as [1], which is the reference you suggested), only the regularization of $I(z; s)$ is considered because they did not treat the sequential VAEs and did not have to care about the mechanism (b).
> Then, to further clarify this point, we have added the discussion about the relation between [1] and our paper to the 3rd paragraph of Section 4 (Update 6).
>
> (ii) Also, the reviewer requires to compare Eq. (13, 14) with alternative approaches since "The merit is thus ... in Eq. (13) and (14)".
> However, as discussed above, we consider that Eq. (13, 14) is one of the instances to regularize Eq. (8), even if Eq. (13, 14) has some merits (this point will be discussed in B1 and B2).
> We have clarified this point by redrafting the 3rd paragraph of Section 3.2 (Update 7).
>
>
> ### A4 Regarding the relation between our paper and [1-3]
> The reviewer queries that our analysis is a "recap" of [1-3].
> We agree that we have missed these important related works;
> however, we believe that our analysis is not the "recap" but is novel due to two reasons.
> (i) Firstly, our analysis clarifies the mechanism (a-c) (see, A1).
> While the phenomenon of (c), i.e., "large $I(x; z)$ results in large $I(z; c=s)$", was previously discussed by [1], the whole mechanism (a-c) has been overlooked in the sequential VAE community.
> (ii) Secondly, by explicitly considering the relationship between the two latent variables $z$ and $s$, our analysis is able to highlight a new problem that was not considered in [1-3].
> For example, [1] considers the relationship between the latent variable $z$ and the observed nuisance factor $c=s$.
> Then, in [1], the focus is only on removing the redundant information from $z$.
> On the other hand, our analysis highlights the need to consider removing the redundant information from $s$ at the same time as removing the redundant information from $z$.
> Although the former has been overlooked in [1-3] and the previous studies for the sequential VAEs, it is an important issue in applications such as voice conversion (see, the 4th paragraph of Section 1).
>
> Also, we thank the reviewer's suggestion that "it would be good to compare against alternative approaches that have the same objective, such as e.g. [1] and [3]".
> We acknowledge that the comparison would improve our paper;
> however, we believe that our regularization term Eq. (8) is novel (see, A3) and different from the objective of [1] and [3].
> Namely, our regularization term is composed of $I(x; z)$ and $I(z; s=c)$.
> On the other hand, the regularization term of [1] is composed of only $I(z; c)$ because [1] did not care about the mechanism (b) (see, also, A3).
> Also, to our understanding, [3] propose a specific network architecture to learn representation.
> Therefore, our proposal is orthogonal to [3] because we propose a regularization term, which is orthogonal to architecture choice (see, A1).
> Moreover, note that, because VQ-VAE2 has a different data generating process from that of DSAE, it cannot perform some applications that DSAE can perform (e.g., voice conversion without speaker labels).
>
> We have clarified the relation between our paper and [1-3] by redrafting the 3rd paragraph of Section 4 (Update 8).

---

> > ### Comment · AnonReviewer5 · 2020-11-23
> > **Answer 2/4**
> >
> > A3: I agree that the precise "combination" of goal/objective and regularizer has not been reported before. But conceptually the problem has been pointed out and addressed before - a couple of times with objectives that look very much like Eq. (8). However, since Eq. (8) is intractable, the most interesting part is in how to make it tractable, which introduces new problems that are solved in different ways (for instance via the DRT in the paper).
> >
> > A4: Thanks for your response to the references mentioned. I agree with most of what you have written in the response, and it is precisely what I wanted to see discussed (in detail!) in the paper. Here's where the detailed novelty of the application and approach lie.

---

> > > ### Author Response · Authors · 2020-11-24
> > > **Author Response (2/3)**
> > >
> > > **A3 "the most interesting part is in how to make it tractable"**
> > > Thank you for your clarification.
> > > We acknowledge that "how to make it tractable" is an interesting part, and have already clarified the merit of DRT compared to alternative choices in B1 and B2.
> > > Also, we have already clarified the difference between Eq. (8) and [1-3] in A4;
> > > then, we would appreciate it if you could suggest another prior methods "that look very much like Eq. (8)", which would be very helpful in clarifying the novel contributions and separating them from what's known.

---

> ### Author Response · Authors · 2020-11-20
> **Author Response to AnonReviewer5 (3/4)**
>
> ## Part B
>
> ### B1 "Comment on ..."
> We thank the reviewer's suggestion and have clarified the following three points;
> however, note that, our claim is not that "Eq. (13, 14) is the best approach" (see A3) even if it has some merit.
>
> (i) "all assumptions made ...".
> As well as prior studies (e.g., Ganin et. al.), the assumption we made is only that the classifier approximates the true density ratio.
> We have added the above comment to the last paragraph of Section 3.2 (Update 9).
>
> (ii) "Are these assumptions justified in practice?"
> This point will be addressed in B2.
>
> (iii) "Would there be alternative ... "
> One of the alternative choices and its downside is discussed in the 3rd paragraph of Section 3.2 of the original manuscript.
> Additionally, the following alternatives can be considered:
> the second term in Eq. (10), which we approximated with a classifier, can also be approximated with other distances such as maximum mean discrepancy, or it can be minimized via Stein variational gradient (see, Zhao et. al.).
> However, a weakness of these methods is that they are difficult to apply efficiently in high dimensions.
> Unfortunately, because the second term treats the random variable $[z, s_1, ..., s_T]$, the dimension size becomes high when $T$ is large.
> We have clarified this point by redrafting the last paragraph of Section 3.2 (Update 10).
>
> ### B2 "Neural network ..."
> We thank the reviewer's suggestion, and have clarified the following two points;
> however, again note that, we use the neural network classifier as an instance to regularize Eq. (8) (see, A3).
>
> (i) "Is the discriminator well calibrated", and "How robust is the method"
> It has been reported by [1] and Iwasawa et. al. that the classifier is not always calibrated.
> However, it is also empirically known that original objectives (in our case, minimizing $D_{\mathrm{KL}} (q(z, s) || p(z)q(s))$) can be reasonably achieved even if the classifier does not perfectly approximate true density ratio (Ganin et. al.).
> In addition, various studies (e.g., Iwasawa et. al. and Miyato et. al.) have proposed techniques to improve the robustness of adversarial training.
> Combining these techniques with our method would potentially improve performance.
>
> (ii) "Is the problem expected to get worse when scaling to ...".
> Adversarial training has been shown to scale when carefully designed (e.g., Brock et. al.).
> Also, the dimension size of the speech data used in our experiment is modestly large (200 x 20), which supports the ability of our method to scale.
>
> We have clarified the above two points by redrafting the last paragraph of Section 3.2 (Update 11).
>
> ### B3 "beta-VAE and ..."
> This point has already been discussed in A4.
>
> ### B4 "Report additional ..."
> We have already incorporated this suggestion in Update 4 (see, A2);
> however, note that, using AoLR and EER falls in line with previous studies in the sequential VAE community (see, A2).
>
> Namely, as expected, CMI-VAE achieved the same level of $I(x; z)$ values and the lower $I(z; s)$ values compared to $\beta$-VAE and MI-VAE, when $\gamma$ becomes large.
> This indicates that CMI-VAE can alleviate the mechanism (c) in A1.
>
> ### B5 "Reporting AoLR ..."
> We thank the reviewer's suggestion, and added the experiments to Appendix K2 (Update 12);
> however, note that, using AoLR falls in line with previous studies in the sequential VAE community (see, A2).
>
> Namely, we found that even if we use a more powerful non-linear classifier (SVM with RBF kernel), our method performed competitively or better than the baselines.
> Here, note that, we use SVM instead of DNN because the dimension size of $x$ and $z$ is not so large and therefore SVM is a proper baseline as well as easy to optimize.
>
> ### B6 "Alemi et al. 2018 gets ..."
> This point has already been clarified in A1 and A2.
> Namely, the reason why we did not perform the evaluation with auto-decoding regime is that our purpose is to learn good global representation.
> We would like to evaluate whether $z$ is good global representation (measured by EER and AoLR), or whether the decoder uses global information within $z$ (measured by mCAS).
> On the other hand, the auto-decoding regime evaluates whether a decoder uses $z$ or not.
>
> ### B7 "Alemi et al. 2018 actually ..."
> We acknowledge that one of their proposals is "a modified objective to target a specific rate";
> however, "$\beta$-VAE as a solution to PC" is also one of their proposals.
> As an evidence of it, the following sentence can be found in Section 5 of Alemi et. al.:
> "We also confirmed that models with expressive decoders can ignore the latent code, and proposed a simple solution to this problem (namely reducing the KL penalty term to $\beta$ < 1)."
> A similar sentence can also be found in Abstract.
> Moreover, He et. al. also employed $\beta$-VAE as a baseline, which supports that $\beta$-VAE is a valid baseline to alleviate PC.
> we have clarified this point by redrafting the 2nd paragraph of Section 2.2 (Update 13).

---

> > ### Comment · AnonReviewer5 · 2020-11-23
> > **Answer 3/4**
> >
> > B1: thanks for the thorough discussion of alternatives!
> >
> > B2: Ideally there would be some control experiments (especially w.r.t. whether the problem gets worse when scaling to higher dimensions / larger application), but thanks for giving a detailed discussion and adding relevant references w.r.t. this to the paper.
> >
> > B3: I still don't think that beta-VAE and MI-VAE are very strong baselines to beat.
> >
> > B4 - B5: Thanks for including this into the paper and clarifying!
> >
> > B6 - B7: I want to point out that the main message and take-away from Alemi et al. 2018 is not "simply use a beta-VAE with beta=1, and all will be fine". The paper proposes something entirely different (targeting a specific rate via the objective), and observes that sometimes a beta-VAE with beta<1 produces solutions that fall into a desirable regime. Please consider the paragraph just above Sec. 3 in Alemi et al. 2018, that even points out a particular theoretical pathology of beta=1. I'm not saying that beta=1 cannot be used as a baseline in the paper, I'm simply saying that using it is only weakly empirically justified by Alemi et al. - but the main message of that paper is a different one.

---

> > > ### Author Response · Authors · 2020-11-24
> > > **Author Response (3/3)**
> > >
> > > **B3 "beta-VAE and MI-VAE are very strong baselines to beat"**
> > > Thank you for your query.
> > > Here we explain why we think that they are the valid baselines.
> > > Firstly, our method is the first to regularize $I(x; z)$ and $I(z; s)$ at the same time (see, A3).
> > > Therefore, we think that a valid baseline should be (i) regularizing only $I(x; z)$, or (ii) regularizing only $I(z; s)$.
> > > The comparison against these baselines can support our claim: not to achieve state-of-the-art results, but to show that regularizing $I(x; z)$ and $I(z; s)$ are complementary (see, A1).
> > > Among them, (ii) would not work well because without regularizing $I(x; z)$, the model suffers from posterior collapse and $z$ has little meaningful information as reported in previous studies (e.g., Alemi et. al.).
> > > Therefore, (i) would be the only valid baseline, and $\beta$-VAE and MI-VAE are the instances of (i).
> > > Secondly, we acknowledge that the comparison against [3] would improve our paper;
> > > however, our proposal is orthogonal to [3].
> > > It is because [3] proposes a specific network architecture while we propose a regularization term (see, A4).
> > > Moreover, note that, because VQ-VAE2 has a different data generating process from that of DSAE, it cannot perform some applications that DSAE can perform (e.g., voice conversion without speaker labels).
> > >
> > >
> > > **B6-B7 "using it is only weakly empirically justified by Alemi et al. - but the main message of that paper is a different one."**
> > > Thank you for your clarification.
> > > Firstly, we acknowledge that "the main message of that paper is a different one" (targeting a specific rate via the objective).
> > > Therefore, we have clarified this point by redrafting the 2nd paragraph of Section 4.
> > > Secondly, however, we believe that $\beta$-VAE is a valid baseline.
> > > It is because $I(x; z)$-maximizing regularization is a standard approach to alleviate posterior collapse, and $\beta$-VAE is a simple and effective instance of the regularization (see, the 2nd paragraph of Section 4).
> > > In fact, the following sentence, which can be found in Section 5 of Alemi et. al., suggests that $\beta$-VAE is a simpler and more effective method for alleviating posterior collapse than the prior methods:
> > > "This fix is much easier to implement than other solutions that have been proposed in the literature, and comes with a clear theoretical justification."
> > > Also, note that $\beta$-VAE has a theoretical justification, which can be found in Section 2 of Alemi et. al, that it can control $I(x; z)$.
> > > That is, since the ELBO contains a positive lower bound and a negative upper bound of $I(x; z)$, the MI can be controlled by balancing the two terms using a weighting parameter $\beta$.

---

> ### Author Response · Authors · 2020-11-20
> **Author Response to AnonReviewer5 (4/4)**
>
> ## Part C
>
> ### C1 "Eq. (10) should be an inequality, because I(z;s) is upper bounded on r.h.s.?"
> Thank you for your query.
> No, Eq (10) is equality because r.h.s. is composed of the upper bound of $I(x; z)$ and the negative upper bound of $I(z; s)$
> The derivation is given in Appendix F.
>
> ### C2 "How was it determined that “alpha=1 works reasonably”, is this based on some control experiments?"
> Thank you for your query.
> "Alpha=1 works reasonably" means that it outperformed the baseline methods ($\beta$-VAE and MI-VAE) in our experiments.
>
> ### C3 "Eq (13). Why this particular mixing in of the KL-term, why not multiply KL(s) with (1-\gamma) as well?"
> Thank you for your query.
> As shown in Eq (13), $L_{SSM} + \gamma I_{CMI-DRT}$ naturally results in the r.h.s..
> Also, note that, reweighting KL(s) results in regularizing $I(x; s)$.
> Since our interest is regularizing $I(x; z)$ and $I(z; s)$ (see, A1), we did not employ the reweighting of KL(s).
>
> ### C4 "Table 1: ..."
> Thank you for your suggestion.
> We have incorporated this suggestion into Appendix K1 and K2 (Update 14).
> Also, we have confirmed that even for large $\gamma$, there is still reasonable reconstruction performance.
>
> ### C5 "Fig 3a - is the x-axis ELBO or KL?"
> Thank you for your query.
> The x-axis is set with ELBO because we expect that some may wonder about the relation between ELBO and AoLR as well as the relation between KL(z) and AoLR.
>
> ### C6 "Fig 3, Table 1: ideally report multiple repetitions with error bars."
> For the experiment using speech data, we have incorporated this suggestion into Appendix K1 (Update 15).
> For the experiment using image data, we are sorry, but we cannot afford to conduct multiple repetitions within the rebuttal period.
>
> ### C7 "For \gamma=0.6 in appendix K, ..."
> Thank you for your suggestion.
> We have clarified this point by redrafting the last paragraph of Section 5.3 (Update 16).
>
>
> Again, thank you for giving us the opportunity to strengthen our manuscript with your valuable comments and queries.
> We look forward to hearing from you regarding our submission.
>
> - Li et. al., Disentangled sequential autoencoder. ICML, 2018.
> - Razavi et. al., Preventing posterior collapse with delta-VAEs. ICLR, 2019.
> - Zhao et. al., Balancing learning and inference in variational autoencoders. AAAI, 2019.
> - Brock et. al., Large scale GAN training for high fidelity natural image synthesis. ICLR, 2019.
> - Iwasawa et. al., Stabilizing Adversarial Invariance Induction from Divergence Minimization Perspective. IJCAI, 2020.
> - Ganin et. al., Domain-adversarial training of neural networks. JMRL, 2016.
> - He et. al., Lagging inference networks and posterior collapse in variational autoencoders. ICLR, 2019.

---

> > ### Comment · AnonReviewer5 · 2020-11-23
> > **Answer 4/4**
> >
> > Thank you for addressing the minor comments. As mentioned previously I will have another thorough read through the paper and the other reviews and then update my review.

---

### Decision · Program_Chairs · 2021-01-07
**Final Decision**

**Decision:**

Reject

**Comment:**

This paper presents a representation method for time series data in the sequential VAE, where the global feature z and local features  s are better disentangled. The intuition behind learning z is to maximize the mutual information between z and input x, while minimizing the mutual information between z and s. The second mutual information is estimated with a discriminator in the DRT framework. Overall, the methodology can be seen as reasonable applications of the disentanglement principle to sequential data. The authors have shown that z and s learned in this way is better disentangled as compared to beta-VAE. In the end, the reviewers feel that while there is good intuitions/technical ingredients, the derivations in Section 3 are not very smooth, and several approximations/choices are not very carefully justified (e.g., choice of alpha, choice of DRT vs. other MI estimators), and perhaps stronger baselines than beta-VAE can be used.

The reviewers rate this paper to be borderline.